# Annotating digital text with phonemic cues to support decoding in struggling readers

**Patrick M. Donnelly** [1,2]*, **Kevin Larson**[3], **Tanya Matskewich**[3], **Jason D. Yeatman**[4,5]

**1** Department of Speech & Hearing Sciences, University of Washington, Seattle, Washington, United States of America, **2** Institute for Learning & Brain Sciences, University of Washington, Seattle, Washington, United States of America, **3** Microsoft Corporation, Redmond, Washington, United States of America, **4** Graduate School of Education, Stanford University, Stanford, California, United States of America, **5** Division of Developmental-Behavioral Pediatrics, Stanford University School of Medicine, Stanford, California, United States of America

\* pdonne@uw.edu

**Data Availability Statement:** All data and analysis code associated with this manuscript are publicly accessible at the following link: [https://github.com/patdonnelly/Donnelly_2019_PLOSONE]

## Abstract

An advantage of digital media is the flexibility to personalize the presentation of text to an individual's needs and embed tools that support pedagogy. The goal of this study was to develop a tablet-based reading tool, grounded in the principles of phonics-based instruction, and determine whether struggling readers could leverage this technology to decode challenging words. The tool presents a small icon below each vowel to represent its sound. Forty struggling child readers were randomly assigned to an intervention or control group to test the efficacy of the phonemic cues. We found that struggling readers could leverage the cues to improve pseudoword decoding: after two weeks of practice, the intervention group showed greater improvement than controls. This study demonstrates the potential of a text annotation, grounded in intervention research, to help children decode novel words. These results highlight the opportunity for educational technologies to support and supplement classroom instruction.

## Introduction

The discrepancy between the value that society places on literacy and reading achievement levels in American youth [1] is a source of concern both among policy makers and scientists [2, 3]. Developmental dyslexia, a learning disability that impacts reading, is widespread, affecting between 5–17% of the population [4, 5]. Beyond dyslexia, poor literacy rates are a nationwide issue, with 34% of fourth graders performing below the Basic level on national achievement tests [6]. Together, these results paint a troubling landscape of literacy achievement and illuminate a non-trivial need for expanding access to evidence-based instruction and intervention.

For many families of struggling readers, access to high quality, evidence-based interventions outside of school are not only limited, but also represent a significant financial burden [7]. Even with a diagnosis, children with reading disabilities struggle to find the support they need in their typical classrooms, necessitating supplemental, after-school programs [8]. For this reason, and with the ever-growing landscape of educational technologies, families are

**Funding:** This work was funded by NSF BCS 1551330, NICHD R21HD092771, Microsoft Research Grants and Jacobs Foundation Research Fellowship to J.D.Y. The funders had no role in study design, data collection and analysis, decision to publish, or preparation of the manuscript. Two authors [K.L. and T.M], are employed by Microsoft Corporation. Microsoft Corporation provided support in the form of salaries for authors [K.L. and T.M.] and research materials but did not have any additional role in the study design, data collection and analysis, decision to publish, or preparation of the manuscript. The specific roles of these authors are articulated in the 'author contributions' section.

**Competing interests:** Two authors [K.L. and T.M.] are employed by Microsoft Corporation. Microsoft corporation only provided financial support in the form of salaries [K.L. and T.M.] and research materials. This does not alter our adherence to PLOS ONE policies on sharing data and materials.

turning to digital alternatives. Many apps and technologies are now widely marketed to augment, or even replace, the teacher in delivering intervention for struggling readers. These new tools are advertised as educational, presented as games, and, in general, are fun to use. Although promising, many of these tools ignore the evidence base on effective instruction and intervention techniques and, of the many educational technologies that are available today, very few have scientific studies testing their efficacy [9–11].

In 2012, it was estimated that hundreds of thousands of educational apps had been released on the Apple iOS app store [10]. According to a RAND report, children between the ages of three and five years-old spend, on average, four hours per day interacting with communications technology (i.e., smart phones, tablets, etc.) [12]. The Joan Ganz Cooney Center reported, based on a survey of parents, that 35% of children aged two to ten years-old use educational apps at least once per week. Furthermore, federal funds are being devoted to bringing Internet and devices to our nation's schools, providing infrastructure for even greater involvement with digital media in education. However, despite the exciting potential offered by educational technology, parents and educators alike feel overwhelmed by the plethora of options and the lack of guidelines surrounding technologies advertised as educational [10].

Decades of scientific research into the behavioral and neural mechanisms of literacy learning has led to the development and testing of effective intervention programs for struggling readers, and established comprehensive guidelines and best-practices for implementation of an effective curriculum [2, 13–15]. Unfortunately, these evidence-based practices (e.g., phonemic awareness, phonics) are largely not being incorporated into the current technological boom. For example, a consistent finding in the intervention literature is that children with dyslexia benefit from direct instruction in phonological awareness and curricula that make clear links between orthography and phonology whereas children with stronger reading skills can often infer grapheme-phoneme correspondences without direct instruction (for review of the extensive literature on the importance of phonics/phonemic awareness see [14, 16–22]). From a sample of 184 apps compiled from online lists of award-winning or highly-rated apps, researchers discovered that although they were entertaining, they lacked scientific backing and "their content, design, production, and distribution are [. . .] an incomplete response to children's literacy needs, especially for struggling readers" [10]. Thus, there is great need for researchers studying literacy development and reading disabilities to work with tech developers on the design of tools that are grounded in the extensive scientific literature on what works for struggling readers, systematically test their effectiveness, and contribute to the development of standards of practice for educational apps targeted at literacy.

Recent metanalyses demonstrate much promise for digital solutions in the context of literacy, yet also describe the multitude of ways that technology is an inappropriate substitute for many aspects of pedagogy [23, 24]. Namely, these meta analyses demonstrate that technologies focused on supplementing what is provided in the evidence-based classroom (i.e. explicit phonics), rather than restructuring at the classroom level, have demonstrated the most promise in the digital landscape. These findings, however, should be interpreted with caution as the authors further contend that the preponderance of studies in this area are characterized by small samples and poor study design [24]. In parallel with research outside of technology, the onus for researchers is to rigorously test digital solutions to discover "what works best in which programs for what students under which conditions" [25].

As a means of scaffolding learning in the classroom, technological tools provide limitless practice [26–28] that can be individualized [24, 29, 30] to optimize for an individual reader's strengths and weaknesses. Additionally, modern computational tools utilizing speech recognition and synthesis application programming interfaces (APIs), allow for embedded tools to be provided in real-time for any given text. One promising avenue for such technology has been

the use of embedded support features such as visual images to facilitate learning sound-symbol correspondence. For example, *Trainertext*, a program in the United Kingdom which provides visual mnemonics above each phoneme in a given text, saw significant improvement after 10 months of exposure and at-home practice [31]. By providing a visual scaffold, the authors concluded that readers could leverage a phonics-based text adaptation to improve their decoding skill. Moreover, *SeeWord Reading app*, a digital tool which uses a picture-embedded font to demonstrate grapheme-phoneme correspondence [32], has demonstrated efficacy in studies performed both in Singapore and the United States [33]. Both *Trainertext* and *SeeWord Reading*, demonstrate the utility of an embedded support to provide concrete visual relationships between written text and spoken language: a foundational skill for literacy. Together these studies demonstrate that technologies focusing at the level of the phoneme and the syllable (or rime) not only benefit reading performance, but also adhere to known learning and pedagogical principles [33, 34].

This paper outlines a collaboration between the Brain Development & Education Lab at the University of Washington and the Learning Tools team at Microsoft to develop *Sound It Out*, a web-based app that annotates text with phonemic image cues to assist in decoding. This tool is the product of collaborative goals to: (1) create an app informed by the literature–in this case, explicit phonics instruction [13], (2) focus on an adaptation over an intervention–a supplemental tool that would assist, not replace the teacher, and allow children to bring skills from the classroom to at-home practice with reading [35–37], (3) design a fun and whimsical interface that children would want to use [25, 38], and (4) enable children to confront their challenges and build the skills to decode more complex words. Our focus on explicit phonics instruction is based on decades of research detailing its importance in literacy education [19–21, 25, 39–41], and the unique role that technology affords to provide limitless exposure and practice inside and outside the classroom. The tool was designed through collaboration between the researchers at the University of Washington (P.M.D. and J.D.Y) and Microsoft (K.L., T.M.), and then tested (independently) in a laboratory study using a pre-registered randomized control trial (RCT) design to determine whether a conceptually simple digital tool can lead to improved word reading outcomes for struggling readers.

As reflected in the preregistered report (available at https://osf.io/q8tpz), the study aimed to answer the following questions: Can struggling readers use phonemic cues to improve reading fluency?, Do struggling readers benefit from phonemic cues when decoding difficult words without time constraints?, and Can we predict which individuals will benefit from the tool based on a standardized battery of reading-related assessments? We hypothesized that the phonemic cue would aid struggling readers in more accurately reading short passages, and with repeated reading, will increase their reading rate. Further, we hypothesized that the phonemic cue would aid struggling readers in more accurately decoding individual real and pseudo-words and that this benefit will relate to the amount of practice they have had with the tool. Finally, regarding individual differences, we hypothesized that those struggling readers that have a specific impairment in phonological processing would benefit the most from the support provided by this tool.

## Methods

### Pre-registration

The methods, including study design, hypotheses, and analysis plan, were pre-registered using the Open Science Framework (OSF) open-access, pre-registration pipeline. We obtained initial reviews and feedback on this pre-registration from an independent OSF reviewer, revised and re-submitted our methodology, and then adhered (with some minor deviations) to this pre-

registered plan throughout the duration of the study. Deviations are noted and explained within this manuscript and are compiled in a 'Transparent Changes' document in the project repository. Documentation is available at https://osf.io/q8tpz.

## Participants

Forty children between the ages of 8 and 12 were recruited from the University of Washington Reading & Dyslexia Research Program database, an online repository of families interested in dyslexia research in the Puget Sound region. The participants (19 females; 21 males) were classified as struggling readers based on a battery of behavioral measurements administered within one year prior to participation in the present study. Here, we use the term "struggling reader" rather than "dyslexia" because there is substantial variability in diagnostic criteria for dyslexia and our goal was to design a tool that would support literacy development for anyone that was struggling, regardless of a dyslexia diagnosis. To be considered a struggling reader, participants needed to have reading skills that were more than one standard deviation (SD) below the mean on either the Woodcock-Johnson IV Tests of Achievement Basic Reading Skills composite (WJ BRS) or the Test of Word Reading Efficiency—2 Index (TOWRE Index), and scores above 1 SD below the mean on the Wechsler Abbreviated Scale of Intelligence Full Scale-2 composite (WASI FS-2). A threshold of 1 SD, rather than the 1.5 SD threshold defined in our preregistration, was adopted to better account for the heterogeneity in the struggling reading population and to expedite participant recruitment. Further, phonological processing abilities were measured for each of the participants using the Comprehensive Test of Phonological Processing– 2 (CTOPP 2) but was not used as an enrollment criterion in the study. Together with age and IQ (WASI-II), CTOPP-2 scores were collected for the purpose of analyzing individual differences in intervention effects to determine if the tool is particularly effective for subjects with certain characteristics. Participants were previously screened for potential speech/language/hearing disorders, neurological impairments, and psychiatric disorders and had none. ADHD was not a disqualifying factor as there is a high co-occurrence with reading disability [42]. In our sample 12 children had a diagnosis of ADHD (6 Control, 6 Intervention). Demographic information on the sample can be found in **Table 1**.

The parents of all participants in the study provided written and informed consent under a protocol that was approved by the University of Washington Institutional Review Board and

**Table 1. Demographic information for study participants.**

| Characteristic | Intervention (N = 20) | Control (N = 20) |
|---|---|---|
| | *Mean (SD)* | *Mean (SD)* |
| **Age (years)** | 10.34 (1.3) | 9.79 (1.1) |
| **Female (proportion)** | 0.5 | 0.45 |
| **WJ Basic Reading Skills Composite** | 78.5 (12.92) | 81.15 (8.43) |
| **TOWRE-2 TWRE Index Composite** | 70.25 (7.48) | 70.65 (7.10) |
| **WASI Full-Scale 2 Composite** | 97.6 (8.75) | 100.15 (17.20) |
| **CTOPP Phonological Awareness Composite** | 87.2 (9.7) | 83.4 (11.94) |
| **CTOPP Rapid Naming Composite** | 79.05 (8.57) | 77.75 (6.7) |

Demographic information for participants in the Intervention and Control groups. See Methods for descriptions of the individual characteristics. For each characteristic, the mean is provided with the standard deviation within parentheses. Independent t-tests—and Wilcoxon signed-rank tests for Age/Gender—demonstrated no significant differences across all characteristics.

all procedures, including recruitment, child assent, and testing, were carried out under the stipulations of the University of Washington Human Subjects Division.

## App design

*Sound It Out* is a web-based application (web app) that annotates text passages with visual phonemic cues to assist decoding. When a passage is viewed using the web app, the vowels appear in blue font with the image cues (located just underneath) indicating the associated phoneme. Each image cue is a highly recognizable symbol whose name contains the target vowel sound for the vowel above. For example, in the word "cow", "o" would appear in blue, with the symbol of a house below. The house cues the child that the letter "o" in "cow" makes the same sound as the /aʊ/ in word "house". **Fig 1** shows three sentences taken from a grade level passage and annotated with *Sound It Out*.

To aid in symbol recognition and retention, the app also integrates a voice cue; when a child presses the phonemic cue symbol, a voice narrates the symbol name followed by an isolated presentation of the target vowel sound. For example, in the "cow" example, when a child presses the image of the house, below "o", a voice will say, "house, /ʊ/". The vowel sounds were recorded by a native English speaker with training in phonetics. The recordings were judged by the three native English-speaking authors to be typical examples of the given vowel sounds and, during the training period, participants were exposed to all the vowel sounds and were able to correctly identify each vowel.

The goal of this app is for the cues to provide helpful hints that aid in decoding and provide children the support they need to attempt to decode difficult words and, eventually, learn the highly inconsistent grapheme-phoneme correspondences of vowels in English. Instead of simply reading challenging words, as is typical in a speech-to-text tool [43, 44], *Sound It Out*

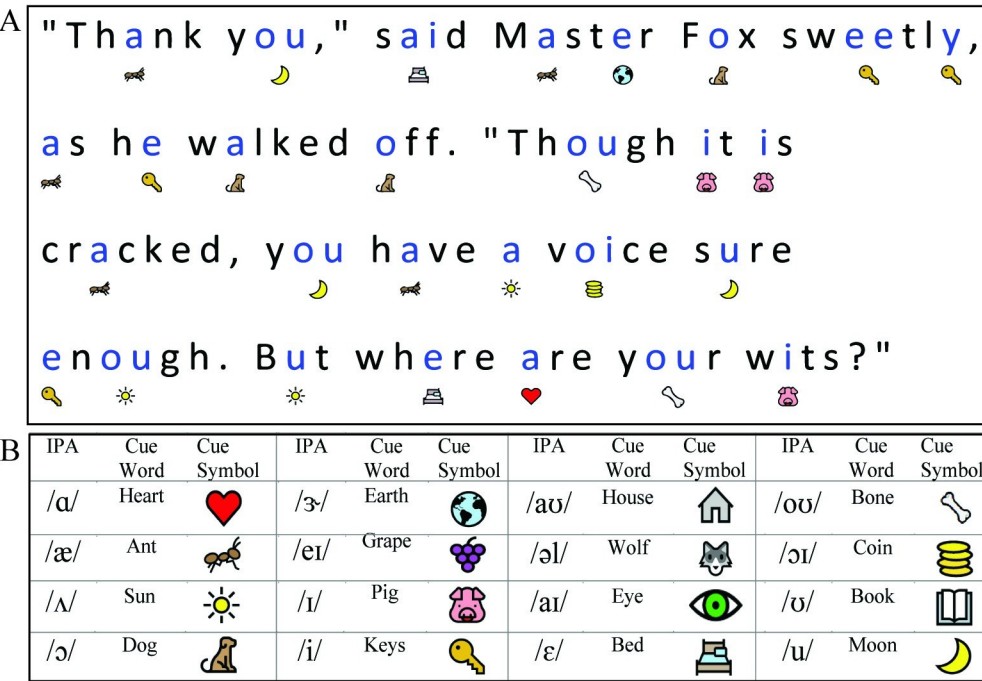

**Fig 1. *Sound It Out* example text.** The vowels in this excerpt from Aesop's 'The Fox and the Crow" fable appear blue with phonemic image cues provided below. The name of the symbols cues the reader to the sound of the target vowel. Reprinted under a CC BY license, with permission from Microsoft Corporation, original copyright 2019.

focuses on a particularly difficult task for struggling readers: vowel decoding. Because vowels in English represent the majority of 'mutually interfering discriminations'—or multiple sound associations - that young readers must master [45], learning the rules for vowels represents a significant portion of phonics curricula [3, 15, 46]. With the understanding that vowels may not be the only challenge for the reader, *Sound It Out* was designed to provide a tool that scaffolds learning and empowers struggling readers to read more complex passages independently. *Sound It Out* was designed as a feature that can be turned on or off when a child is reading. In this study, both intervention and control participants were given a tablet and taught how to use it for reading. The primary manipulation was whether the *Sound It Out* feature was turned on. Beyond this manipulation, the text in the table-based reader was identical.

## Procedure

**Study design.**   In a randomized pre-post design, participants were randomized to a control or intervention condition. Randomization was unconstrained with group assignment determined at time of consent; however, sibling participants were assigned to the same group to better control participant adherence. Both intervention and control participants completed an initial, baseline session that collected all outcome measures using the normal text condition presented on a Kindle fire tablet without the *Sound It Out* cue (see **Outcome Measures**). Control participants completed a brief training in use of the tablet then a two-week, at-home practice program without the *Sound It Out* tool. Intervention participants completed the full training period that included both the use of the tablet as well as a formalized introduction to and practice with the *Sound It Out* cue prior to an identical at-home practice program with the tool turned on. After two weeks, all participants returned for a post-intervention session where the intervention participants were tested with the cue, while control participants experienced the normal text condition for all study stimuli. As the goal of this study was to test proof-of-concept for a digitally embedded phonetic cue in a small scale RCT, generalization to un-cued reading in the intervention group was not tested.

**Training program.**   For the intervention group, each child was first oriented to the presence of the cue in an example passage. The researcher would show a passage with the cue and say the following: "Now we are going to use a cool tool that we made to help you read the tricky words. Underneath each word we have symbols that help you figure out the sound that the blue letters make." The researcher would walk through a word and demonstrate how the cues could be used to help decode words. Then the researcher explained when to use the cues: "When you come to a word you don't know, just look at the symbols and that should help you figure out the sounds that the blue letters make." The child was then instructed to read through the passage. When they came to tricky words the researcher alternated between clicking on the symbols and naming the symbols to help sound out the words that cause difficulty. If the child read through the first example passage with ease, a more challenging passage was added to ensure that the child had demonstrated efficient and correct usage. Then, with the use of flashcards, the researcher reviewed each symbol explicitly with the child.

For the control group, participants were introduced to the tablet and instructed on how to navigate to the various passages for at home practice. They were not shown the phonemic cues, but otherwise followed an identical procedure.

*At-home practice*. After training with the application, and when the baseline testing session was completed, participants were provided tablets to take home for reading practice. Participants were asked to read at least one story per day over the course of two weeks using the app (at home). For the intervention group, the *Sound It Out* feature was enabled so that phonemic cues showed up in the passages. For the control group, text was rendered in the same font but

without the cues. Tablets were pre-loaded with 36 first, second, and third grade supplementary passages for children to read with or without their parents. To encourage meaningful practice, parents were provided with a brief introduction to the app prior to taking home the tablet and were given instructions to allow the child to work through difficult words using the image cues prior to providing any additional guidance. For those that adhered to these instructions, and assuming each passage would take approximately ten minutes to complete, each child experienced at least 100 minutes of exposure over the two-week practice period. Adherence to the practice schedule was measured via short, three-question comprehension quizzes completed after reading a story (through a web interface). A participant was only credited for a practice passage if they received a comprehension score of at least two.

Supplementary passages for at-home reading were from ReadWorks.org, an online library of grade-level passages (used with permission of ReadWorks). Passages were phonetically coded manually by the research team based on the most common pronunciation in the Pacific Northwest dialect of English. The code was verified by both P.M.D & K.L., with inconsistencies discussed and decided via consensus. All passages were displayed on an Amazon Kindle Fire 8 tablet at a set font size and resolution. Comprehension questions, to gauge practice adherence, were created by a trained, certified teacher at a local school that specializes in working with children with developmental dyslexia and were grade-level matched to the individual passages. Passages and comprehension questions can be found in the supplementary material (see **S1 File**).

**Outcome measures.** Measures of reading performance were collected at baseline and after the two-week period of practice. Real and pseudo word decoding accuracy was measured by having participants read lists of words that were loaded into the web app. Four unique lists of 30 real and 30 pseudo words were created with two lists being delivered at each session. All lists were developed using the orthographic wordform database MCWord [47]. Real word lists consisted of the most frequent words in the English language with five instances each of three-letter to eight-letter words. Pseudo word lists consisted of the most frequent bigrams in English with an identical progression from three-letter to eight-letter pseudo words. (See **S2 and S3 Tables** for detailed word statistics). All lists were unique and were rated for consistent difficulty using timed reading in ten typically reading adults. Lists were administered in a counterbalanced order for participants in each group. Lists were phonetically coded manually by the first author based on the most common pronunciation in the Pacific Northwest dialect of English. Accuracy in pronunciation was not limited to the code used for the phonemic cues but extended to acceptable pronunciations in English. At the start of administration, all participants were reminded that they were not being timed and encouraged to read as accurately as possible. For intervention participants exposed to the image cue, participants were additionally reminded that the symbols were there to help them should they come to a challenging word. Post-hoc analysis revealed that performance was highly reliable across the different word lists: performance was highly correlated for the two lists of real words ($r = 0.86$, $p < 0.001$) and pseudo words ($r = 0.79$, $p < 0.001$) in each session. Accuracy of real and pseudo word decoding was our primary outcome measure (number of words read correctly on each list akin to the Woodcock Johnson Word ID and Word Attack (both untimed measures)).

Passage reading rate and accuracy was measured by having participants read grade-level passages that were loaded into the web app. All testing passages used were from the Dynamic Indicators of Basic Early Literacy Skills (DIBELS) 6th Edition library, used with permission of the University of Oregon Center on Teaching & Learning. These passages are commonly used as benchmark assessments in schools and have been extensively used in reading research. Only passages rated at second, third, and fourth grade were used. For each testing session, every passage was presented by a research assistant in the Brain Development & Education Lab and

audio recorded for accurate scoring and coding. As with the decoding measures, participants were reminded at the start of administration that they were not being timed, encouraged to read as accurately as possible, and (for intervention participants) that the symbols were there should they come to a challenging word. Instead of constraining the oral reading to the one-minute limit of the DIBELS protocol, all passages were read to completion twice in a repeated reading design. Each passage reading yielded four measures: accuracy (number of words pronounced correctly) on the first and second read; and rate of reading (number of accurate words per minute) on the first and second read. To test the influence of *Sound It Out* on connected text reading, analyses focused on the word-reading accuracy of the first read and the word-reading rate of the second read. Second reading rate was used to control for the added time that might be associated with using the phonemic cues to sound out difficult words on the first read. Testing passages were coded and rated in the same manner as the at-home practice passages.

**Statistics.** Due to the presence of missing data, data were analyzed with linear mixed effects (LME) models, as specified in our pre-registration. Missing data consisted of word reading accuracy and rate information for twelve passages from seven participants due to testing fatigue or inability to complete the passages. For each outcome measure, we fit an LME model with fixed effects of: (1) time (pre-intervention / post-intervention as a categorical variable); (2) group (intervention / control groups as a categorical variable); (3) the group by time interaction. The models included a random intercept for participant, to account for individual variation in baseline performance. To account for differences between the individual, lab-created word lists, we added a random intercept for word list to those models. Practice data were used to ensure that all participants engaged with the tool at home and were also used in correlational analyses to examine the impact of at-home exposure on improvement.

Due to issues collecting reliable usage statistics for the at home reading practice, prediction analyses were not appropriate. Instead, exploratory correlation analyses were performed using the Pearson correlation coefficient between post-pre difference scores and the three subject characteristics collected at baseline: age, WASI-II and the CTOPP-2. This analysis differs from that described in the preregistration due to the small number of reading variables collected and inability to collect robust measures of exposure, making methods of dimensionality reduction not appropriate. Analyses were carried out using the NumPy and SciPy libraries of Python and the MATLAB Statistics Toolbox (2019a) [48]. All data and analysis code associated with this manuscript are publicly accessible at the following link: [https://github.com/patdonnelly/Donnelly_2019_PLOSONE]

## Results

### Phonemic cues improve decoding accuracy for real and pseudo words

For our primary outcome measure (as specified in our pre-registration [link]), children were assessed on their ability to decode lists of increasingly more complex real and pseudo words prior to, and immediately following, the two week intervention period. For this measure of decoding accuracy, words were displayed in a list (**Fig 2A and 2B**). **Fig 2** also includes bar plots of difference scores as well as violin plots of the full score distribution.

For real-word decoding accuracy, although both groups did show some growth, the group by time interaction was not significant ($\beta = 1.3$, $t(156) = 1.923$, $p = 0.056$) indicating that the growth in the intervention group was not statistically different from the control group.

For pseudo-word decoding accuracy the group by time interaction was significant ($\beta = 3.175$, $t(156) = 2.99$, $p = 0.003$) with the intervention group showing significantly greater improvement than the control group (a threshold of 0.0125 was defined in the preregistered

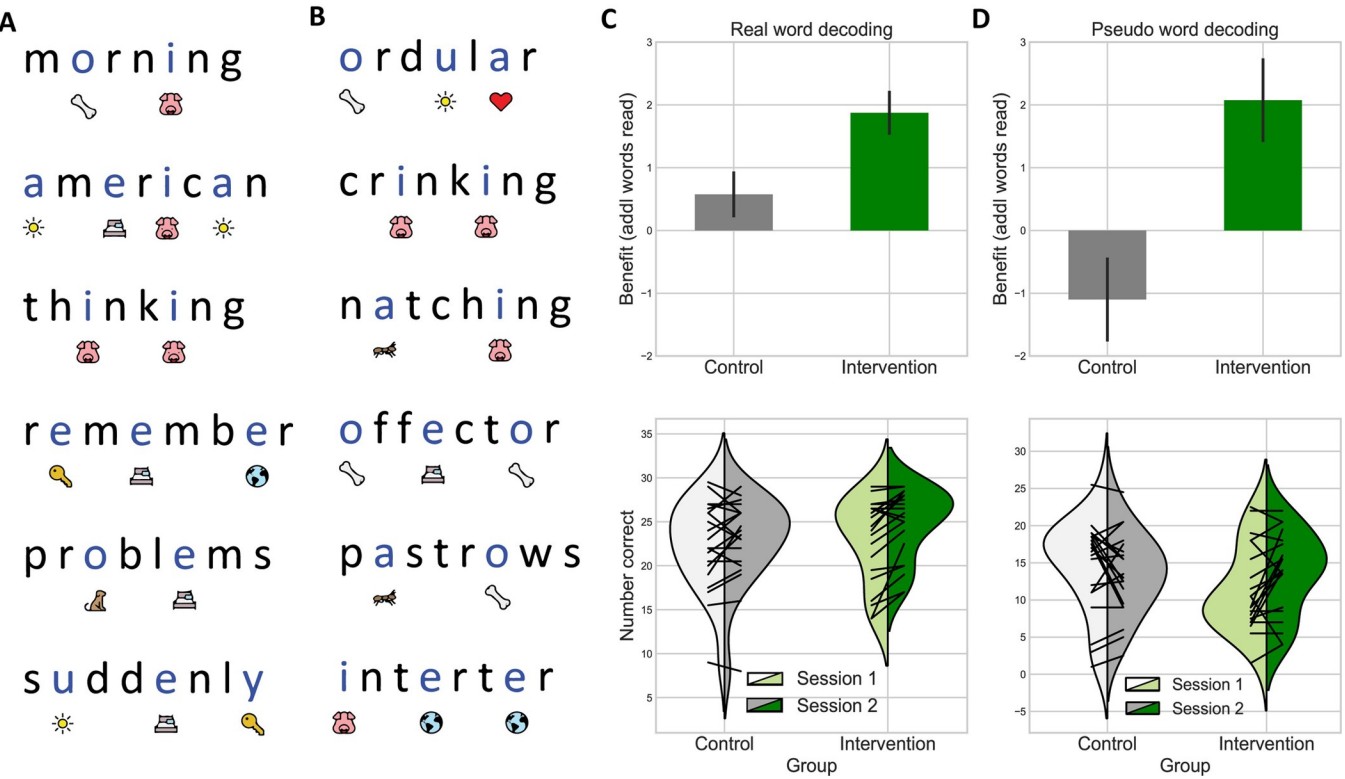

**Fig 2. Untimed decoding performance on real and pseudo words.** Example stimuli from the real-word (A) and the pseudo-word (B) lists with Sound it Out phonemic cues below the highlighted vowels. Bar plots show difference scores (number of words read correctly) from the first and second sessions for both the control and intervention groups on real word (C) and pseudo word (D) lists. Bar heights represent the additional words read on the second session. Error bars reflect +/- 1 SEM. Below the bar plots, violin density plots show group performance on these measures for both sessions with superimposed line plots of individual performance.

report to adjust for multiple comparisons). At pretest, despite randomization, the intervention group by chance had lower scores than the control group: this is evidenced by the significant main effect of group in the mixed effects model ($\beta$ = -6.2, t(156) = -2.65, p = 0.009).

To determine the influence of practice on individual outcomes, we examined the correlation between at-home practice and individual growth. All participants completed at-home practice (intervention Mean = 13 stories, SD = 6; control Mean = 12, SD = 6), but the correlation between amount of practice and growth was not significant for the intervention group (real words r = -0.14, p = 0.55; pseudo words r = 0.23, p = 0.35) or the control group (real words r = 0.01, p = 0.97; pseudo word r = -0.37, p = 0.10).

Regarding our final question on subject characteristics that predict individual response, a correlation analysis revealed no significant relationships between our baseline predictors (age, WASI-II Full Scale-2, and CTOPP-2) and improvement in real-word decoding. For pseudo-word decoding, there were negative correlations with the CTOPP-2 Phonological Awareness (PA) (r = -0.52, p = 0.018) and Phonological Memory (PM) (r = -0.48, p = 0.034) composite measures as well as the WASI-II Full Scale– 2 score (r = -0.49, p = 0.027), but these effects were not significant after correcting for multiple comparisons. Due to the heterogeneity of our sample, we tested a model with added covariates for age and initial phonological awareness ability: model fit comparison revealed no benefit to the more complex model and no significant main effects for the added covariates (**S1 File**).

These findings show that without the constraints of time during testing, there was a beneficial effect of access to the phonemic cue for single word decoding. This benefit was observed in the case of pseudoword decoding where children were asked to pronounce novel words in isolation. Moreover, correlation analyses suggest that this effect is more pronounced for those participants with more significant impairments in phonological processing and lower IQ (though these effects did not surpass our adjusted significance threshold of $p < 0.0125$). Although the effect sizes were moderate (Cohen's $d = 0.74$ for pseudoword decoding, $d = 0.57$ for real word decoding), these results suggest that children, with practice, can incorporate a novel cue to scaffold independent decoding.

## Phonemic cues for connected text reading

To determine whether the phonemic cue confers a benefit for reading connected text, we assessed word reading accuracy and rate on grade level passages before and after intervention. **Fig 3** depicts bar plots of difference scores and violin density plots for the control and intervention participants in terms of (a) reading accuracy: number of words read correctly in the passage on the first read; and (b) reading rate: number of correct words per minute in the second read.

For word reading accuracy the group by time interaction was not significant ($\beta = 0.014$, $t(65) = 1.1$, $p = 0.275$). For word reading rate there was a non-significant group by time interaction ($\beta = 0.014$, $t(64) = 0.368$, $p = 0.714$). Effect sizes were $d = 0.36$ for accuracy and 0.13 for rate. To examine the effect of heterogeneity in our sample, we tested a model with added covariates for age and initial phonological awareness ability: model fit comparison and analysis of added fixed effects demonstrated no significant effects of these covariates (**S1 File**).

Correlation analyses revealed only a near-significant negative relationship between age and word reading accuracy ($r = -0.55$, $p = 0.028$) suggesting that younger children may benefit more from *Sound it Out*.

## Discussion

Using a small scale RCT design, we tested the hypotheses that struggling readers could leverage a phonemic image cue placed below the vowels in digitally presented text to improve reading accuracy for isolated words and connected text, and that this benefit would be more pronounced for those readers with lower performance on measures on phonological processing. Data collected after a two-week period of unsupervised (but digitally monitored) practice demonstrated that struggling readers could read more complex words using the tool: compared to the control group, the intervention group showed a significantly larger improvement in decoding accuracy specifically for pseudo-words. As depicted in the results, this benefit did not extend to either measure of connected-text reading (accuracy and rate) and the improvement in real-word decoding did not differ significantly between intervention and control groups.

Although there was no benefit, stable performance on measures of connected text reading was observed for all participants with no significant difference between groups. The lack of benefits for connected text reading might reflect the limited training period or the increased cognitive demands of a novel approach to reading. These are important questions for future studies as generalization to connected text is of key importance.

Correlation analyses, after multiple comparison correction, revealed no significant relationships between our variables of interest and benefit of the cue. Due to unreliable practice data (see Statistics), analyses cannot support any conclusions regarding the relationship between subject characteristics and benefits conferred by phonemic cues. However, results suggest that the tool may benefit those participants who are younger and/or have lower phonological

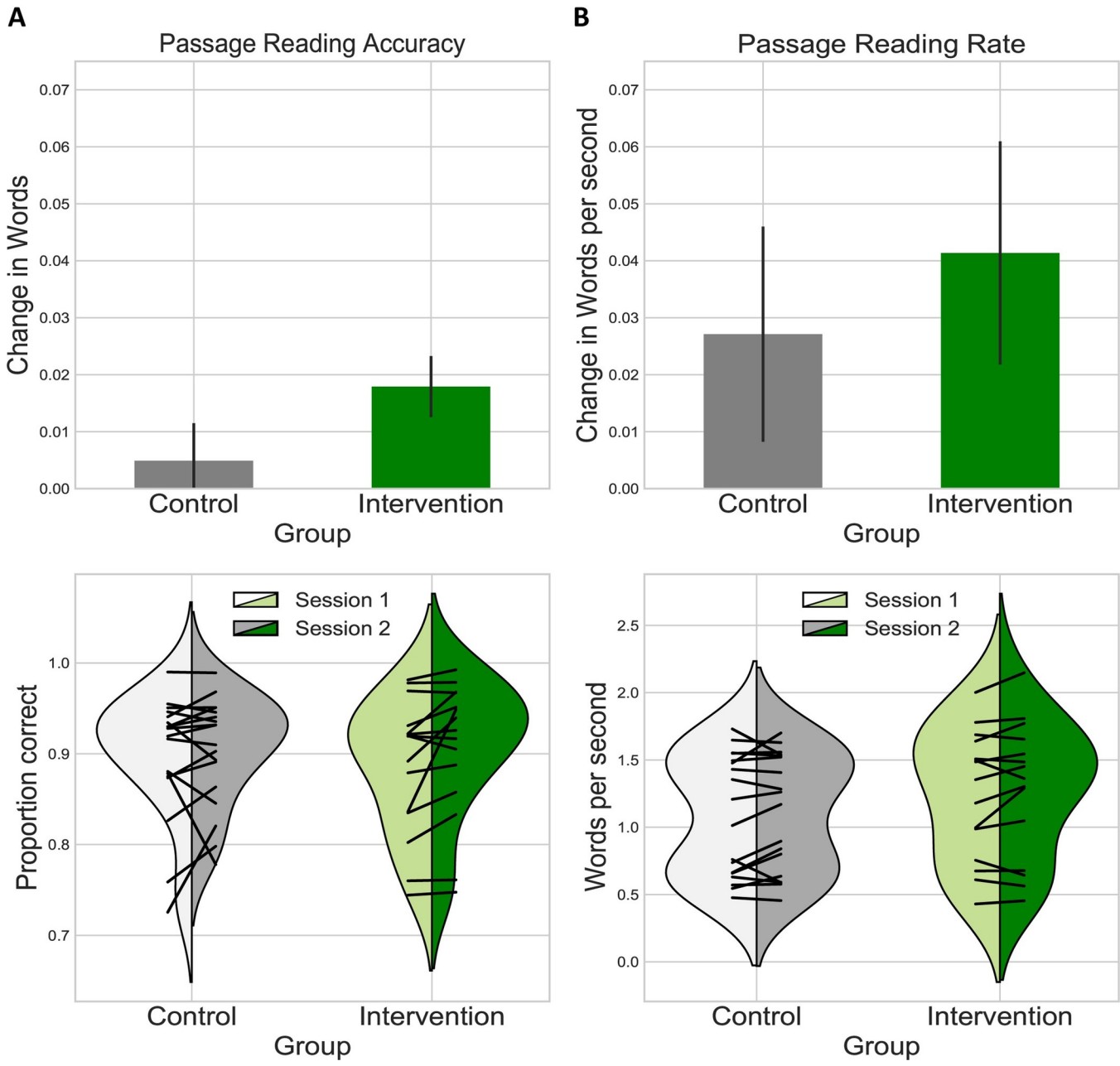

**Fig 3. Accuracy and rate for connected text reading.** Bar graphs depict difference scores from the first and second sessions for both the control and intervention groups for word reading accuracy (A) and rate (B). Bar heights represent the additional number of words read correctly and additional accurate words per second on the second session. Error bars reflect +/- 1 SEM. Below the bar plots, violin density plots show group performance on these measures for both sessions with superimposed line plots of individual performance.

processing scores (see Results). Together, although most analyses failed to meet our adjusted significance threshold, data suggests that participants were able to effectively use the cues in isolated situations (i.e. pseudoword reading), but the tool did not become sufficiently automatic to produce significant gains in passage reading fluency.

Decades of dyslexia research has been devoted to developing and systematically testing interventions designed for struggling readers. In the digital age, devices provide ever-

expanding access to a plethora of educational apps and resources advertised as educational. Many families seek out these resources to supplement their child's education. Unfortunately, most of this educational technology lacks scientific backing [49]. A goal of this project was to embed a core feature of evidence-based practice in literacy instruction into a digital tool to scaffold learning to read. Inspired by the key tenants of phonics instruction (e.g., explicit and clear instruction in letter-sound correspondence, repeated exposure, and systematic practice), the phonemic cue was designed to provide struggling readers with a hint to aid the decoding of novel words. We focused on vowels because, in English, the highly inconsistent grapheme-phoneme mapping is a major hurdle for struggling readers. In a landscape of digital tools that provide instant corrections at the whole word level, the phonemic cue annotation in this study is one of a only a few learning aids that provides an element of instruction (at the phoneme level) to support generalizable skill [31, 33, 43, 50]. Instead of being given the answer at the first sign of struggle, the child can utilize the phonemic cue to learn part of the word, yet still needs to exercise the building blocks of literacy to get the answer.

In a similar vein to this work, researchers in the UK developed *Trainertext*, a scaffolding program that provides whimsical, visual mnemonics for scaffolding letter-sound correspondence. Instead of focusing on vowels, *Trainertext* provides a visual cue above every phoneme in connected text that is associated with short rhyme. For example, above the word "gas", *Trainertext* has images related to the phrases "Goat in a Boat", "The Ant in Pink Pants", and "The Snake with a Shake" to represent the grapheme-phoneme correspondence of each letter. A RCT with individualized instruction and 10 months of exposure demonstrated significant improvements (Cohen's d = ~0.80) with the largest effects seen for decoding and phonological awareness (Messer & Nash, 2018). This work demonstrated how phonics-based text annotation could be leveraged by struggling readers to bootstrap their decoding skills, and that, with extended exposure, that benefit could generalize to decoding without the cues. The present study built on this work by (a) employing a simplified symbol set focused on vowel sounds, with the hope of building a tool that would be quicker to learn and less cognitively demanding and (b) could be used immediately without requiring months of a resource-intensive intervention program. Taken together, these two studies emphasize that text annotation is a promising approach, either in combination with an in-person intervention (as in [31]), or as a tool to support at-home practice (as in the present study).

As was the case with *Trainertext*, *Sound It Out* requires children to learn a new, albeit intuitively designed, symbol system and practice sufficiently for the associations to become automatic. The symbols chosen were optimized for recognizability, but the challenge remains in teaching the child to associate a portion of the symbol name with a discrete sound segment in an often-unrelated word. Moreover, the sound-symbol association must be fast enough to not impede short term memory with increased cognitive load [51–53]. These two dimensions, effective use and automaticity, are captured by the two areas of measurement: the untimed word lists and the connected text reading.

The finding of improved pseudoword decoding performance indicates that without temporal constraints or the cognitive demands of connected text reading, children may be able to use the phonemic cue to improve decoding performance. This suggests that a brief, two-week practice period was enough and the tool sufficiently intuitive to have an impact. As the limited effects in passage reading accuracy and rate reveal, however, the tool did not extend to situations when time constraints were re-introduced. Either due to the limited practice period, limited supervised practice, or conflict with existing strategies children use when approaching challenging words, children did not similarly benefit from the phonemic cues in connected text. Future studies should incorporate qualitative and metacognitive methods to identify factors and circumstances that encourage struggling readers to adopt a novel strategy.

Albeit promising, these results should be interpreted cautiously: Our power analysis indicated that we only had sufficient power to detect relatively large effects and many of the analyses (e.g., individual differences) were likely underpowered. Also, as there was a significant difference at pre-test for our sole finding with pseudo word decoding, future studies are needed to rule out the role of regression toward the mean and possible ceiling effects. Thus, future work is needed with larger sample sizes to provide more conclusive results. Moreover, two additional points merit further investigation. First, future experiments should more efficiently, and quantitatively and qualitatively, monitor practice adherence and cadence at home to better explore the relationship between exposure and reading-related measures. Second, given the short intervention period, we did not examine generalization to reading improvements without the cue and across different aspects of skilled reading. We only investigated whether the cue could be effectively used to decode more complex words. Thus, examining long-term learning effects and generalization to a variety of different contexts, as well as the role of parental involvement/participation is an important future direction.

We had anticipated that the cue would require limited exposure, but our results are in stark contrast with previous studies that have instituted a more comprehensive, extended training program and observed significant benefit to fluent reading [31, 54]. Thus, although the training and practice periods were enough to encourage effective use, they were not enough to ensure automatic and fluent use in a natural setting, and by extension, were insufficient to make general claims on efficacy of *Sound it Out* for supporting long-term growth in reading skills.

On the other hand, the limited effects in accuracy and rate performance suggest that either the cue did not adversely impact reading performance or that it was underutilized given the increased cognitive demands of real-time reading. Many participants in the study have received supplemental instruction previously and have learned strategies for approaching new words. As a novel strategy, the phonemic cue may have been overridden when children were asked to read continuously and for comprehension. In line with the corpus of research on strategy instruction for literacy [39, 51, 55, 56], although the children in our sample demonstrated competency in the use of the cue in isolated, single-word decoding, strategy adoption would require more sustained exposure.

A strength of educational technologies for literacy is their ability to empower parents, teachers and other advocates to support and supplement their child's learning outside the classroom [31, 57–60]. *Sound it Out* is unique in that it provides a tool that gives parents a strategy for reinforcing phonics principles with their child. Many parents, when confronted with the stress and challenge of raising a child who struggles with learning to read, are told to read more to their child, but not given the knowledge base needed to provide meaningful support [61]. Post-study feedback from parents in the study were overwhelmingly positive, with a majority of parents noting interest in using the tool into the future (see **S1 Table**). Relatedly, a study by Ronimus and colleagues demonstrated increased efficacy of *GraphoGame*, a digital literacy program, for reading performance when children engaged with the tool with parental involvement [57]. Thus, with parental support and potential alignment with the teacher and in-school curriculum [24, 49, 62–66], *Sound it Out* represents a promising venture bridging research and practice.

In aggregate, these findings represent a small scale *proof-of-concept* for this novel approach to assisting struggling readers by merging the extensive evidence base on effective literacy instruction and the affordances that technology lends to the educational arena. Not only did it prove promising in improving decoding performance with very limited practice, but it also was observed to be non-detrimental to passage reading, meaning that it was not too cognitively demanding. Future research should focus on optimizing training and practice to produce

gains that will extend beyond isolated single word decoding and lead to more confident, fluent readers.

## Supporting information

**S1 Table. Parent/child responses to post-study questionnaire for the intervention group.** Listed are the responses to the post-study questionnaire for the intervention group participants and their parent. After completing the study, children were asked to answer honestly to the following questions: Did you like the app? And would you like to use the app again in the future? Parents were then asked if they enjoyed using the app. Those adults who did not respond did not participate in the practice to comment on the app.
(DOCX)

**S2 Table. Real word frequency statistics.** Frequency information for real word stimuli. Words were retrieved from MCWord Orthographic Wordform Database ([http://www.neuro.mcw.edu/mcword/](http://www.neuro.mcw.edu/mcword/)). According to the database, the frequency is a measure of how often the wordform occurred in 1,000,000 presentations in the CELEX database.
(DOCX)

**S3 Table. Pseudoword frequency statistics.** Frequency information for pseudoword stimuli. Pseudowords were retrieved from MCWord Orthographic Wordform Database ([http://www.neuro.mcw.edu/mcword/](http://www.neuro.mcw.edu/mcword/)). According to the database, the constrained bigram frequency is a measure of how often the bigram wordform occurred in 1,000,000 presentations in the CELEX database.
(DOCX)

**S1 File. Additional analyses, at-home practice passages and comprehension questions.** Each passage is provided as well as the comprehensions that were used to determine passage completion. Additional analyses, with explanations, are also provided.
(PDF)

## Acknowledgments

We would like to thank Greg Hitchcock and both the Advanced Reading Technologies and the Learning Tools teams for their aid in the design and development of *Sound It Out*. This was an ideal collaboration between research and industry and would not have been possible without the Microsoft team's support and commitment to the pursuit of science. We would also like to thank Taylor Madsen and the members of the Brain Development & Education Lab at the University of Washington for their input, feedback, and support.

## Author Contributions

**Conceptualization:** Patrick M. Donnelly, Kevin Larson, Tanya Matskewich, Jason D. Yeatman.

**Data curation:** Patrick M. Donnelly.

**Formal analysis:** Patrick M. Donnelly.

**Funding acquisition:** Jason D. Yeatman.

**Investigation:** Patrick M. Donnelly.

**Methodology:** Patrick M. Donnelly, Jason D. Yeatman.

**Project administration:** Patrick M. Donnelly, Jason D. Yeatman.

**Resources:** Patrick M. Donnelly, Kevin Larson, Jason D. Yeatman.

**Software:** Patrick M. Donnelly, Kevin Larson, Tanya Matskewich.

**Supervision:** Patrick M. Donnelly, Jason D. Yeatman.

**Validation:** Patrick M. Donnelly.

**Visualization:** Patrick M. Donnelly.

**Writing – original draft:** Patrick M. Donnelly.

**Writing – review & editing:** Patrick M. Donnelly, Jason D. Yeatman.

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
