## [Decision Letter · Decision Letter 0]

26 May 2020

PONE-D-19-34461

Annotating digital text with phonemic cues to support decoding in struggling readers

PLOS ONE

Dear Dr. Donnelly,

Thank you for submitting your manuscript to PLOS ONE. After careful consideration, we feel that it has merit but does not fully meet PLOS ONE’s publication criteria as it currently stands. Therefore, we invite you to submit a revised version of the manuscript that addresses the points raised during the review process.

As you can see from the detailed comments of both reviewers (which I approve from my reading of the manuscript), they above all request a more comprehensive review of existing studies on embedded digital support tools and a clearer motivation of changes from the preregistration and particular methodological decisions.   

We look forward to receiving your revised manuscript.

Kind regards,

Claudia Männel, PhD

Academic Editor

PLOS ONE

Journal Requirements:

"This work was funded by NSF BCS 1551330, NICHD R21HD092771, Microsoft

Research Grants and Jacobs Foundation Research Fellowship to J.D.Y. The funders

had no role in study design, data collection and analysis, decision to publish, or

preparation of the manuscript"

We note that one or more of the authors are employed by a commercial company: "Microsoft Corporation"

Reviewers' comments:

Reviewer's Responses to Questions

**Comments to the Author**

1. Is the manuscript technically sound, and do the data support the conclusions?

Reviewer #1: Partly

Reviewer #2: No

2. Has the statistical analysis been performed appropriately and rigorously? 

Reviewer #1: I Don't Know

Reviewer #2: No

3. Have the authors made all data underlying the findings in their manuscript fully available?

Reviewer #1: Yes

Reviewer #2: Yes

4. Is the manuscript presented in an intelligible fashion and written in standard English?

Reviewer #1: Yes

Reviewer #2: Yes

5. Review Comments to the Author

Reviewer #1: There is very limited literature on Embedded Support Features, which can provide critical supports to struggling readers for understanding new content and participating in learning activities. Therefore, this study, which explores the efficacy of a tablet-based text annotation reading tool in intervention research, to help children decode novel words, fills a gap in the literature on embedded digital supports. The study makes a convincing case that while the edtech market has grown tremendously as new apps are being released every year and federal funds are being devoted to provide infrastructure for even greater involvement with digital media in schools, many of the edtech tools “lacked scientific backing”.

While the study rightfully argues that edtech tools should go beyond just being entertaining and should be backed by research-based best practices, it fails to provide some literature on the best-practices. The current study lacks important and needed detail regarding how the existing literature would be evaluated to arrive at meaningful conclusions. For example, the study should refer to the extensive scientific literature on what works for struggling readers in terms of best practices as well as existing studies on embedded digital support tools. A review of the existing literature on digital supports would have provided a better understanding of what is known--which digital features or supports have been found to have promise to support struggling students’ learning and under what conditions. The study made some references to such tools/studies in the Discussion section, which should have been presented in the Literature Review section.

I agree with the authors who argue that “there is great need for researchers studying literacy development and reading disabilities to work with tech developers on the design of tools that are grounded in the extensive scientific on what works for struggling readers, systematically test their effectiveness, and contribute to the development of standards of practice for educational apps targeted at literacy”. However, even though we have “evidence-based practices (e.g., phonemic awareness, phonics)”, we cannot assume that the best practices identified in non-digital contexts may easily apply or be expanded within digital learning contexts. The study makes such an assumption.

The study states that “Intervention participants completed an extended training period” but it is not clear what “extended” means.

One of the weaknesses of the study is to conclude that 100 minutes of exposure over the two-week practice period is enough to make conclusions about the efficacy of a tablet-based text annotation reading tool. The result could simply be due to the novelty effects, which pose a threat to external validity because they make it difficult to know if the results of the study are due to a treatment that works or due to the novelty of a treatment.

Reviewer #2: The authors describe a small-scale randomized control trial with 40 struggling readers aged 8-12 who are split across two intervention groups. The aim was to investigate the effect of annotating text with phonemic cues in a tablet app by allowing children to play for a two-week intervention phase. Their results indicate that there is a benefit on untimed pseudoword decoding accuracy, but no effects on word decoding accuracy or connected text reading accuracy and speed.

The manuscript is well written, the experimental design was preregistered and all data was made available. There are some shortcomings in this study, some of which relate to the experimental setup itself, and others to unjustified deviations from the study preregistration.

Major:

1) The manuscript itself does not contain any research questions and hypothesis and only refers to the preregistration on OSF. This should be changed and implemented into the manuscript. Furthermore, the preregistration contains an additional question relating to individual phonological skills which is not touched upon in the manuscript.

2) As far as I understand from the manuscript the word lists and reading passages at post-training assessment were carried out with the phonemic cues present as depicted in Figure 2. It remains unclear whether this feature was only activated for the intervention group (1) or both groups (2). For 1) this would mean that what is being measured is the effect of cueing but not the generalization of the training on un-cued reading. Option 2) would mean that control children are confronted with this annotation for the first time at an assessment moment, which would increase their cognitive load and put them at a disadvantage. In either case, the impact of training should be assessed without cueing in a real-world reading scenario.

3) P13:L21: "a threshold of 0.025 was defined in the preregistered report to adjust for multiple comparisons". The report contains a threshold of 0.0125 which would render the sole significant finding not significant. Where does this deviation come from?

Minor:

Introduction:

Adding phonemic cues in form of symbols above text is not necessarily bound to digital media per se and could equally be done on paper. It remains unclear why there is a focus on implementing this inside an app. How does this relate to previous literacy intervention work? Has this been done on paper and were there comparable findings?

Method:

P6:L17: The preregistration report mentions 1.5SD below the mean as an inclusion criterion for being a struggling reader, here it is 1SD. Where does this deviation come from?

The group characteristics provided in Table 1 do not contain p-values to show that the groups do not differ from one another. Could you also explain the randomization/matching used for group assignment? Were there any constraints so that the groups would end up being comparable?

In the discussion you mention that many participants were receiving supplemental instruction for their reading difficulties. Can you specify how many and how their distribution among the two groups were? Potential gains may ultimately also stem from simultaneous speech therapy if the groups are not balanced.

Given that your sample is quite heterogenous with an age range of 8-12 years and the intention to include a broad range of struggling readers it might be worthwhile to add additional co-variates to your models to support a more causal and generalizable interpretation of your intervention (e.g. for age or pre-test phonological awareness skills - which was also an unmentioned research question).

Could you provide additional information on the custom word lists you created? E.g. frequency information for the words and neighborhood density for the pseudowords. Furthermore, reliability information would also be relevant, i.e. Cronbach's alpha, split-half reliability or correlation between lists.

Matlab script: Is there a specific reason why the models where fit with method 'ML'? This is only useful for stepwise model comparison and final models for publication should be fitted with REML as this gives a less biased estimation and more robustness towards outliers.

Results:

You mention missing data. Could you specify which variables from how many subjects are affected by this?

I much appreciate access to the raw data and code! It would be nice if you added a code book indicating what measure each variable contains.

P14:L3: Looking at the numbers of played stories (mean=13/12, SD=6) it seems that intervention fidelity was not very high. Could you provide more information on that? How many minutes of training would this translate to? What was the range of minimum and maximum exposure? Did children adhered to the instruction of playing one story per day? What happened to children that played less? Did they start playing each day and then stop after a few days? Were there children that did not play for a week and then played 10 session on the last day?

P14:L3: Instead of looking into correlations I would suggest including app exposure as a covariate into your models.

P14:L10: Thank you for providing an effect size. It would be informative if you could also provide these for all other discussed effects.

The preregistration report states: "Based on pilot data it is reasonable to expect that the intervention will help subjects read 5-10 additional words on the untimed word lists." as well as "subjects differ, on average, by +/- 3 words on repeated administrations of the word lists". Based on these numbers a statistical power of 0.9 was estimated.

In the actual study the results are much lower: 1 additional word and 3 additional pseudowords, which falls into the range of the expected test-retest reliability you mentioned. Which makes me wonder how robust is this effect? The actual power will be much lower and the reported effect size of d=0.74 seems way too large for such a short intervention period. I would be very careful at overinterpreting these findings as they are probably not generalizable.

Discussion:

Given the comments above, possible effects should be discussed with more caution. It also remains unclear what was actually measured with the experimental design (see major point 2 above). This should be made clearer when discussing potential effects.

The lack of research questions in the manuscript makes the discussion rather unspecific. Here you should get back to your initial questions and hypothesis.

P15:L11: "In connected text reading, growth was observed for all participants but without a significant difference between groups." These effects are essentially zero (0.01 - 0.04 words per minute extra), so they don’t have a practical relevance.

P16:L16: There are many different effect size measures which differ in their scales. Please mention which one is meant.

I'm missing a limitations section where you discuss possible lack of power, adherence to daily app use and the need for replication in a bigger sample.

Summary:

In sum, I agree that text annotation is a promising approach, but it seems that this study describes null results. Those are still relevant to make publicly available and I would recommend to further clarify the experimental design and conduct an exploratory analysis by adding relevant co-variates to the models (pre-test phonological awareness skills, age, app exposure).

6. PLOS authors have the option to publish the peer review history of their article (what does this mean?). If published, this will include your full peer review and any attached files.

Reviewer #1: No

Reviewer #2: Yes: Toivo Glatz

---

## [Author Response · Author response to Decision Letter 0]

9 Jul 2020

A detailed response to reviewers has been uploaded with this submission. We humbly thank the reviewers for their thoughtful comments and critiques that we believe have led to a stronger manuscript overall.

---

## [Decision Letter · Decision Letter 1]

18 Aug 2020

PONE-D-19-34461R1

Annotating digital text with phonemic cues to support decoding in struggling readers

PLOS ONE

Dear Dr. Donnelly,

Thank you for submitting your manuscript to PLOS ONE. After careful consideration, we feel that it has merit but does not fully meet PLOS ONE’s publication criteria as it currently stands. Therefore, we invite you to submit a revised version of the manuscript that addresses the points raised during the review process.

We look forward to receiving your revised manuscript.

Kind regards,

Claudia Männel, PhD

Academic Editor

PLOS ONE

Reviewers' comments:

Reviewer's Responses to Questions

**Comments to the Author**

1. If the authors have adequately addressed your comments raised in a previous round of review and you feel that this manuscript is now acceptable for publication, you may indicate that here to bypass the “Comments to the Author” section, enter your conflict of interest statement in the “Confidential to Editor” section, and submit your "Accept" recommendation.

Reviewer #1: (No Response)

Reviewer #2: (No Response)

2. Is the manuscript technically sound, and do the data support the conclusions?

Reviewer #1: (No Response)

Reviewer #2: Partly

3. Has the statistical analysis been performed appropriately and rigorously? 

Reviewer #1: (No Response)

Reviewer #2: No

4. Have the authors made all data underlying the findings in their manuscript fully available?

Reviewer #1: Yes

Reviewer #2: Yes

5. Is the manuscript presented in an intelligible fashion and written in standard English?

Reviewer #1: Yes

Reviewer #2: Yes

6. Review Comments to the Author

Reviewer #1: I would like to thank the authors for the changes they have made in this version.

The authors made an effort to address my comments in the previous round of review. For example, I commented that the study should refer to the extensive scientific literature on what works for struggling readers in terms of best practices as well as existing studies on embedded digital support tools. While this version provides some literature review of existing studies on embedded digital supports, it still doesn't provide any literature review on research-based best practices that the authors refer to such as (e.g., phonemic awareness, phonics etc.).

Additionally, in this version, they referred to 3 meta-analysis (9, 16, 17) that demonstrate promise for digital solutions in the context of literacy. However, all these 3 studies were conducted pretty much by the same authors. It looks as if the authors published the same or similar meta-analyses in three different publications. If that is the case, I would advise referring to only one of the meta-analysis rather than all three and cite some other existing studies on embedded digital support tools. Rather than explaining each study, synthesize results into a summary of what is and is not known, identifying areas of controversy in the literature.

Reviewer #2: Review for PONE-D-19-34461R1: "Annotating digital text with phonemic cues to support decoding in struggling readers"

I would like to thank the authors for transparently addressing the raised points and providing a much-improved manuscript. Below is a list of mostly minor points which still require further attention or clarification. My only major concern relates to the data analysis and provided code.

Introduction:

Regarding research questions (P6:L14-17) and hypotheses (P6:L17-23) none of the research questions contains exposure and you added the hypothesis of an intervention x exposure interaction to the second research question, although it would also be valid for the first one, and ultimately perhaps be a better fit with the third one. Is there a specific reason for that?

Method:

Following up on your clarification of my mayor point 2 in the previous version of the manuscript, you added that at post-intervention only intervention participants saw the cues and control participants had cues disabled. Could you further clarify how the assessment looked at pre-test? Did both groups read without cues? Or was it already enabled for the intervention group? And did the assessment take place before or after the introduction training with the app, or was the pre-assessment done offline in a pencil and paper style?

Considering that grapheme-phoneme associations also depends on availability of high-quality phoneme representations, could you clarify whether the recordings you added to the app were judged for prototypicality, and whether children were instructed to play with headphones or not?

P7:L16: I assume classification as struggling reader happened within one year "prior" to participation?

P8:L11: If ADHD was not an exclusion criterion could you provide numbers on how many children had this diagnosis?

P11:L22: The instruction "When you come a word you don't know..." does not appear to be grammatical?

P13:L1-3: Given that you do not have logs for exposure and credit exposure based on the comprehension questions I wanted to point out that the comprehension questions are all True/False responses and therefore the probability of getting 2 or more correct replies by just guessing (and therefore being credited with the exposure) is rather high at 0.5. Did you consider to analyze these comprehension data for intervention effects as well?

P14:L7: Please avoid exponential notation for p-values and shorten to p < 0.001 according to common convention.

P14:L9: Could you clarify whether the Woodcock Johnson test measures timed or untimed decoding?

Data analysis:

P8:L13: In the caption of Table 1 you say that you conducted independent t-tests. However, not all variables are normally distributed, and you must use the wilcox/mann-whitney test in these cases. It does not influence the fact that there are not differences between the groups though.

Thank you for carrying out additional analyses with age and initial phonological awareness as covariates. Could you still add these models to the supplementary information and add a sentence in the results section indicating that this was done and yielded comparable results?

P15:L6-L15: Thank you for providing the code book which allowed me to have a look at the data myself. There are still mayor issues with the data analysis though. In the manuscript you state that you added independent random intercepts of time and participant - which would suggest (1|time) and (1|participant), in the Matlab code you have session nested within participants (1-session|participant) as well as a reading list random intercept (1|acc_Indicator). The former is not well specified though, because "1-" ignores the session part. It should be specified as (1+session|participant) or because the 1 is implied (session|participant). Was there a specific intention behind using "1-"? In any case, this changes the coefficients only minimally but changes the confidence intervals quite a bit. I'm also concerned that this is a very complex random effects structure for the little data you have and you might be overfitting. I would suggest to do model comparison (with AIC/BIC) to see which random effects structure is required. Did you also check that model assumptions were fulfilled after fitting (normality and homoscedasticity of residuals?)

Furthermore, the statistics which are presented in the results section do only partially stem from the provided Matlab code. It is, for example, not apparent where the main effects of group and session (P16:L21-L22) come from and why they only have 78 degrees of freedom when these effects have 156 DF in the model. Please make sure to provide all code for results which you provide in the manuscript (also for possible post-hoc or correlation analyses as well as calculations of effect sizes).

With the pseudoword model (P17:L3-L8) there is also the issue that it describes a big and significant difference between the two groups at pretest (with the intervention group scoring 6 words lower) while at post-test the two groups are at the same level again. This is not mentioned in the results section and sheds a different light on the sole significant effect.

This is made worse by plotting only pre-post differences and the use of bar charts to represent the data as it distorts the perception of observed values and draws attention to unimportant aspects (i.e. bar height rather than difference between means). See, e.g.: https://doi.org/10.1371/journal.pbio.1002128

Consider using dotplots or boxplots of the raw pre-post data with an indicator of the means.

P16:L17: I understand that due to unreliable usage statistics the correlation analysis was the best you can do, but this is not a valid approach for a prediction analysis. It's best to be transparent about the unreliable data by adding a few sentences from the response letter to the statistics part of the methods section and also bring this up in the limitations. Or remove this analysis altogether after mentioning that this part of the data collection did not work as intended.

Results:

P16:L21: You are using ß (latin sharp s) instead of β (greek beta), as well as a p-value with 5 decimals.

P18:L1-3: Repeated use of "pronounced". Furthermore, the effect was not "particularly pronounced" for the pseudoword decoding, but it was exclusively there.

P18:L3-5: Please present the statistics if you refer to effects which suggest something. Also in the Matlab script.

P19:L3: Significance and effect sizes are completely unrelated. We can observe giant yet non-significant effects, as well as extremely small (and thus irrelevant) yet highly significant effects.

Discussion:

P19:L22-P20:L4: I find this part a bit misleading as it jumps back and forth: things are inconclusive; a relation between pretest phonological awareness and intervention outcome is mentioned for which no statistic is provided in the results; two mentions that after p-value adjustment there were no effects left, but it is still discussed that these effects might suggest something. Especially at the start of the discussion there should be a clear summary of what was found first and then one can discuss what the presence and absence of effects might indicate.

P20:L16: [...] highly inconsistent grapheme-phoneme mapping "of English" [...]

P22:L3-L5: Throughout the manuscript there are sentences which use many commas and therefore appear encapsulated and make it unnecessarily hard for the reader to grasp the most relevant information. In this case you can straight out write that there is an improved decoding of pseudowords and remove the subordinate clause. As this is the only comment regarding writing style, I also wanted to note that there appear to be a lot of double spaces in manuscript. Content wise it could be highlighted here that this improvement appeared after being trained with the app for (only) 2 weeks.

P23:L5-L13: This is an interesting observation. They appeared to have learned to make use of the cues, but this disappears in the testing situation. I feel it’s quite relevant to understand this better. Given that there is no performance penalty it seems like they might be able to ignore the cues altogether? One could take a more qualitative approach and investigate how children use it in these different scenarios or alternatively add an assessment mode into the app itself which blends with the intervention?

P23:L23: Given that parents also got a brief introduction to the app and you come back to it in the discussion it might be worth to recommend measuring parental involvement in future studies?

Summary:

In summary, the manuscript has much improved and will be a relevant addition to the field, but the provided analysis and code are not yet up to the required standards.

7. PLOS authors have the option to publish the peer review history of their article (what does this mean?). If published, this will include your full peer review and any attached files.

Reviewer #1: No

Reviewer #2: **Yes: **Toivo Glatz

---

## [Author Response · Author response to Decision Letter 1]

25 Sep 2020

We would like to thank the reviewers once again for their thoughtful and insightful comments on this study. We have made further revisions and agree with the reviewers’ points that these changes will make this manuscript suitable for publishing. The reviewers comments are below in black and our responses and specific actions are noted in blue.

Reviewer #1: I would like to thank the authors for the changes they have made in this version.

The authors made an effort to address my comments in the previous round of review. For example, I commented that the study should refer to the extensive scientific literature on what works for struggling readers in terms of best practices as well as existing studies on embedded digital support tools. While this version provides some literature review of existing studies on embedded digital supports, it still doesn't provide any literature review on research-based best practices that the authors refer to such as (e.g., phonemic awareness, phonics etc.).

We thank the reviewer for this added clarification on the missing references in our literature review section. We have added necessary citations to the statement above “ (e.g., phonemic awareness, phonics etc.)” on page 4. Beyond explanation of our phonemic cue in the context of other embedded supports, we believe a more thorough review of specific instructional techniques for phonics/phonemic awareness is beyond the scope of this manuscript. In this revision we do, however, refer the reader to the most comprehensive review papers and meta-analyses. Additionally, explanation of the focus of the app is provided in the App design section (page 9-10) detailing its use in the context of broader phonics curricula. 

The section of the introduction now reads:

“Decades of scientific research into the behavioral and neural mechanisms of literacy learning has led to the development and testing of effective intervention programs for struggling readers, and established comprehensive guidelines and best-practices for implementation of an effective curriculum [2,13–15]. Unfortunately, these evidence-based practices (e.g., phonemic awareness, phonics) are largely not being incorporated into the current technological boom. For example, a consistent finding in the intervention literature is that children with dyslexia benefit from direct instruction in phonological awareness and curricula that make clear links between orthography and phonology whereas children with stronger reading skills can often infer grapheme-phoneme correspondences without direct instruction (for review of the extensive literature on the importance of phonics/phonemic awareness see [16–23]). ” (P4, L8-17)

Additionally, in this version, they referred to 3 meta-analysis (9, 16, 17) that demonstrate promise for digital solutions in the context of literacy. However, all these 3 studies were conducted pretty much by the same authors. It looks as if the authors published the same or similar meta-analyses in three different publications. If that is the case, I would advise referring to only one of the meta-analysis rather than all three and cite some other existing studies on embedded digital support tools. Rather than explaining each study, synthesize results into a summary of what is and is not known, identifying areas of controversy in the literature.

Although the meta analyses cited have very similar authorship, they provide slightly different perspectives on the literature that we believe provide a better picture of the state of research in this area. However, we do agree that the first (Cheung & Slavin, 2011) substantially overlaps with the Slavin, et al., 2011 and have removed the first from the citation. The primary difference in contribution between the Slavin, et al., 2011 and the Cheung & Slavin, 2013 is that the latter provides a specific context for struggling readers whereas the former applies to all elementary school youth. We believe that both perspectives are highly relevant and have amended the section to the following with some added synthesis: 

“Recent metanalyses demonstrate much promise for digital solutions in the context of literacy, yet also describe the multitude of ways that technology is an inappropriate substitute for many aspects of pedagogy [24,25]. Namely, these meta analyses demonstrate that technologies focused on supplementing what is provided in the evidence-based classroom (i.e. explicit phonics), rather than restructuring at the classroom level, have demonstrated the most promise in the digital landscape. These findings, however, should be interpreted with caution as the authors further contend that the preponderance of studies in this area are characterized by small samples and poor study design [25]. ” (P5, L3-10)

Reviewer #2: Review for PONE-D-19-34461R1: "Annotating digital text with phonemic cues to support decoding in struggling readers"

I would like to thank the authors for transparently addressing the raised points and providing a much-improved manuscript. Below is a list of mostly minor points which still require further attention or clarification. My only major concern relates to the data analysis and provided code.

Thank you

Introduction:

Regarding research questions (P6:L14-17) and hypotheses (P6:L17-23) none of the research questions contains exposure and you added the hypothesis of an intervention x exposure interaction to the second research question, although it would also be valid for the first one, and ultimately perhaps be a better fit with the third one. Is there a specific reason for that?

We thank the reviewer for identifying this point. We indicated an exposure effect in the second question only because we believed that children would require significantly more practice than the 2-week period to motivate effective use during connected text reading (question 1). As the second question related to single word decoding in a more “assessment” context we believed we would see more active use and, in extension, more effect of exposure. 

Method:

Following up on your clarification of my mayor point 2 in the previous version of the manuscript, you added that at post-intervention only intervention participants saw the cues and control participants had cues disabled. Could you further clarify how the assessment looked at pre-test? Did both groups read without cues? Or was it already enabled for the intervention group? And did the assessment take place before or after the introduction training with the app, or was the pre-assessment done offline in a pencil and paper style?

We apologize for the confusion that was met in parsing our study design. The first sentence of the study design section indicates that “all participants completed an initial, baseline session that collected all outcome measures using a normal text condition presented on a Kindle fire tablet without the Sound It Out cue” (L6-8). The training period for both groups (intervention and control) occurred after this baseline testing with the normal text condition as detailed in the remaining paragraph. All testing was completed on a Kindle Fire tablet. We have further adjusted the text to clarify this point.

Considering that grapheme-phoneme associations also depends on availability of high-quality phoneme representations, could you clarify whether the recordings you added to the app were judged for prototypicality, and whether children were instructed to play with headphones or not?

The recordings were created by a member of the study team (KL) and were judged to be typical examples of the vowel sounds by PD & JD. All three are native English speakers. Furthermore, the vowel sounds were played during the training period and kids were given the opportunity to indicate if a given recording was unclear. The following has been added to the manuscript for clarity on this point: 

“The vowel sounds were recorded by a native English speaker with training in phonetics. The recordings were judged by the three native English-speaking authors to be typical examples of the given vowel sounds and, during the training period, participants were exposed to all the vowel sounds and were able to correctly identify the vowels.” (P10, L20-P11,L2)

P7:L16: I assume classification as struggling reader happened within one year "prior" to participation?

Yes - the statement has been amended to the following: “The participants (19 females; 21 males) were classified as struggling readers based on a battery of behavioral measurements administered within one year prior to participation in the present study.” (P8, L4-6)

P8:L11: If ADHD was not an exclusion criterion could you provide numbers on how many children had this diagnosis?

Yes - the following statement has been added: “In our sample 12 children had a diagnosis of ADHD (6 Control, 6 Intervention).” (P9, L1-2)

P11:L22: The instruction "When you come a word you don't know..." does not appear to be grammatical?

Thank you for catching this typo - we have amended the statement to the following: “When you come to a word you don’t know, just look at the symbols and that should help you figure out the sounds that the blue letters make.” (P12, L19-21)

P13:L1-3: Given that you do not have logs for exposure and credit exposure based on the comprehension questions I wanted to point out that the comprehension questions are all True/False responses and therefore the probability of getting 2 or more correct replies by just guessing (and therefore being credited with the exposure) is rather high at 0.5. Did you consider to analyze these comprehension data for intervention effects as well?

We did look to see if there were any differences in practice between control and intervention participants, and found that there was no difference between the distributions of practice using an independent t test (t = -0.592, p=0.558). 

P14:L7: Please avoid exponential notation for p-values and shorten to p < 0.001 according to common convention.

We have amended where applicable. 

P14:L9: Could you clarify whether the Woodcock Johnson test measures timed or untimed decoding?

We have added clarification - the statement now reads: “Accuracy of real and pseudo word decoding was our primary outcome measure (number of words read correctly on each list akin to the Woodcock Johnson Word ID and Word Attack (both untimed measures)).” (P15, L4-6)

Data analysis:

P8:L13: In the caption of Table 1 you say that you conducted independent t-tests. However, not all variables are normally distributed, and you must use the wilcox/mann-whitney test in these cases. It does not influence the fact that there are not differences between the groups though.

Thank you for pointing this out - I have performed the Wilcoxon signed-rank test on Age/Gender to conclude that all characteristics demonstrate no significant differences. The caption now reads: 

“Demographic information for participants in the Intervention and Control groups. See Methods for descriptions of the individual characteristics. For each characteristic, the mean is provided with the standard deviation within parentheses. Independent t-tests - and Wilcoxon signed-rank tests for Age/Gender - performed demonstrate no significant differences across all characteristics.”

Thank you for carrying out additional analyses with age and initial phonological awareness as covariates. Could you still add these models to the supplementary information and add a sentence in the results section indicating that this was done and yielded comparable results?

We have added the following to the manuscript to indicate that this analysis was performed with a parenthetical reference to a more in-depth explanation in the supplement:

“Due to the heterogeneity of our sample, we tested a model with added covariates for age and initial phonological awareness ability: model fit comparison revealed no benefit to the more complex model and no significant main effects for the added covariates (S1 File).” (P18, L23 - P19, L3)

“Similarly concerned with the heterogeneity of our sample, we tested a model with added covariates for age and initial phonological awareness ability: model fit comparison and analysis of added fixed effects demonstrated no significant effect (S1 File).” (P20, L10-13)

The following has been added to the Supplement:

“Role of Age and Phonological Awareness as Covariates to Mixed Effects Models

 Due to the heterogeneity of our struggling reader population in both reading ability and age, we performed an exploratory analysis of these results testing a model that adds covariates for participant age and initial phonological awareness ability (as measured using the CTOPP phonological awareness composite measure).

Model fits were compared using AIC/BIC values and revealed that in no case was the model with added covariates a superior model to the simpler model used in the manuscript. Looking at the results of these models, moreover, revealed no significant effects of age or phonological awareness. Interaction effects were unchanged from the simpler models reported in the manuscript. 

The model fit workflow and code associated with this analysis can be found in the associated project repository on GitHub.”

P15:L6-L15: Thank you for providing the code book which allowed me to have a look at the data myself. There are still mayor issues with the data analysis though. In the manuscript you state that you added independent random intercepts of time and participant - which would suggest (1|time) and (1|participant), in the Matlab code you have session nested within participants (1-session|participant) as well as a reading list random intercept (1|acc_Indicator). The former is not well specified though, because "1-" ignores the session part. It should be specified as (1+session|participant) or because the 1 is implied (session|participant). Was there a specific intention behind using "1-"? In any case, this changes the coefficients only minimally but changes the confidence intervals quite a bit. I'm also concerned that this is a very complex random effects structure for the little data you have and you might be overfitting. I would suggest to do model comparison (with AIC/BIC) to see which random effects structure is required. Did you also check that model assumptions were fulfilled after fitting (normality and homoscedasticity of residuals?)

We thank the reviewer for drawing attention to this issue in our statistical model. We had in fact run model comparisons to determine the ideal random effects structure. The notation issue raised by the reviewer was due to incorrect documentation in a previous version of MATLAB. The best fitting model was, in fact, the simple one suggested by the reviewer with a random intercept for participant. We now provide a more detailed model fit workflow. For the word list data, we also added a random intercept for list number (1|acc_Indicator) because we found it to vastly improve model fit due to slight variations between word lists (though including or excluding this random effect does not impact the effects of interest). We have repeated all analyses using this random effects structure with no major changes to our main findings. We also add to our uploaded code checks for normality in residual distribution/heteroscedasticity. Analyses reveal that in all model fits there is a normal distribution of residuals and no evidence of heteroscedasticity. 

The following has been amended to the manuscript:

“For each outcome measure, we fit an LME model with fixed effects of: (1) time (pre-intervention / post-intervention as a categorical variable); (2) group (intervention / control groups as a categorical variable); (3) the group by time interaction. The models included a random effect for participant, to account for individual variation in baseline performance. To account for differences between the individual, lab-created word lists, we added a random effect for word list to those models. ” (P16, L6-11)

We have also amended the “Transparent Changes” document in our preregistration. 

Furthermore, the statistics which are presented in the results section do only partially stem from the provided Matlab code. It is, for example, not apparent where the main effects of group and session (P16:L21-L22) come from and why they only have 78 degrees of freedom when these effects have 156 DF in the model. Please make sure to provide all code for results which you provide in the manuscript (also for possible post-hoc or correlation analyses as well as calculations of effect sizes).

The mentioned analyses were performed post-hoc, and for clarity have been changed to paired t-tests on subsets of the dataset to help illustrate the differences between groups and provide perspective for the interaction results. Discussion of these results in the manuscript have been amended to clarify that distinction: 

“For real-word decoding accuracy the group by time interaction was not significant (β=1.3, t(156) =1.923, p=0.056) indicating that the growth in the intervention group was not statistically different from the control group. Post-hoc, paired t-test analyses dividing the data at the group level revealed a significant increase in in the intervention group (t(19) = 3.75, p = 0.001) and a non-significant increase in the control group (t(19) = 1.10, p = 0.285).”

“For pseudo-word decoding accuracy the group by time interaction was significant (β=3.175, t(156)=2.99, p=0.003) with the intervention group showing significantly greater improvement than the control group (a threshold of 0.0125 was defined in the preregistered report to adjust for multiple comparisons). At pretest, despite randomization, the intervention group by chance had lower scores than the control group: this is evidenced by the significant main effect of group in the mixed effects model. Post-hoc, paired t-test analyses at the group level revealed there was a significant increase in the intervention group (t(19) = 2.176, p = 0.042) and a small but non-significant decrease in the control group (t(19) = -1.149, p = 0.265).”

“For word reading accuracy the group by time interaction was not significant (β=0.014, t(65)=1.1, p=0.275). Post-hoc, paired t-test analyses dividing the data at the group level revealed a non-significant increase in the intervention group (t(15) = 2.325, p = 0.035) and a non-significant increase in the control group (t(17)=0.518, p = 0.611). [JY1] For word reading rate there was a non-significant group by time interaction (β=0.014, t(64)=0.368, p=0.714). Post-hoc, paired t-test analyses at the group level revealed a non-significant increase in both the intervention (t(14)=1.468, p = 0.164) and control groups (t(17)=1.000, p = 0.331).”

We have also ensured that these and other analyses are available in the public repository. 

With the pseudoword model (P17:L3-L8) there is also the issue that it describes a big and significant difference between the two groups at pretest (with the intervention group scoring 6 words lower) while at post-test the two groups are at the same level again. This is not mentioned in the results section and sheds a different light on the sole significant effect.

This is made worse by plotting only pre-post differences and the use of bar charts to represent the data as it distorts the perception of observed values and draws attention to unimportant aspects (i.e. bar height rather than difference between means). See, e.g.: https://doi.org/10.1371/journal.pbio.1002128

Consider using dotplots or boxplots of the raw pre-post data with an indicator of the means.

This is a very important point and we have added the following to the results section to add clarity and transparency:

“For pseudo-word decoding accuracy the group by time interaction was significant (β=3.175, t(156)=2.99, p=0.003) with the intervention group showing significantly greater improvement than the control group (a threshold of 0.0125 was defined in the preregistered report to adjust for multiple comparisons). At pretest, despite randomization, the intervention group by chance had lower scores than the control group: this is evidenced by the significant main effect of group in the mixed effects model. Post-hoc, paired t-test analyses at the group level revealed there was a significant increase in the intervention group (t(19) = 2.176, p = 0.042) and a small but non-significant decrease in the control group (t(19) = -1.149, p = 0.265).” (P18, L3-10)

We have also added violin plots to the Supplemental information (S1 Fig) to provide more detailed visualizations of the data. 

P16:L17: I understand that due to unreliable usage statistics the correlation analysis was the best you can do, but this is not a valid approach for a prediction analysis. It's best to be transparent about the unreliable data by adding a few sentences from the response letter to the statistics part of the methods section and also bring this up in the limitations. Or remove this analysis altogether after mentioning that this part of the data collection did not work as intended.

Thank you for this insight. We agree that it is important to be more transparent about this limitation and have amended the statistics section to read:

“Due to issues collecting reliable usage statistics for the at home reading practice, prediction analyses were not appropriate. Instead, post-hoc correlation analyses were performed using the Pearson correlation coefficient between post-pre difference scores and the three subject characteristics collected at baseline: age, WASI-II and the CTOPP-2. This analysis differs from that described in the preregistration due to the small number of reading variables collected and inability to collect robust measures of exposure, making methods of dimensionality reduction not appropriate.” (P16, L14-20)

We have also amended the Discussion section to specify Correlation rather than Prediction analysis and draw the reader’s attention again to the limited nature of this dataset:

“Correlation analyses, moreover, were inconclusive but suggested that the tool may benefit those participants who are younger and/or have lower phonological processing scores (see Results). However, these results were not significant (after multiple comparison correction) and, due to unreliable practice data (see Statistics), insufficient to support any conclusions regarding the relationship between subject characteristics and benefits conferred by phonemic cues” (P21, L11-16)

We have also added the following to the limitations section:

“First, future studies should more efficiently, and quantitatively and qualitatively, monitor practice adherence and cadence at home to better explore the relationship between exposure and reading-related measures.” (P24, L7-9) 

Results:

P16:L21: You are using ß (latin sharp s) instead of β (greek beta), as well as a p-value with 5 decimals.

We have amended accordingly. 

P18:L1-3: Repeated use of "pronounced". Furthermore, the effect was not "particularly pronounced" for the pseudoword decoding, but it was exclusively there.

We have amended the paragraph to read: 

“These findings show that without the constraints of time during testing, there was a beneficial effect of access to the phonemic cue for single word decoding. This benefit was observed in the case of pseudoword decoding where children were asked to pronounce novel words in isolation. Moreover, prediction analyses suggest that this effect is more pronounced for those participants with more significant impairments in phonological processing and lower IQ (though these effects did not surpass our adjusted significance threshold of p < 0.0125). “ 

P18:L3-5: Please present the statistics if you refer to effects which suggest something. Also in the Matlab script.

Now when we say suggest, we provide relevant statistics that suggest this idea. In cases that we do not have a statistic, but suggestion of a hypothesis in future work, we note that in the paper.

P19:L3: Significance and effect sizes are completely unrelated. We can observe giant yet non-significant effects, as well as extremely small (and thus irrelevant) yet highly significant effects.

We have amended the statement to read: “Further, data reflected effect sizes of d = 0.36 for accuracy and 0.26 for rate.”

Discussion:

P19:L22-P20:L4: I find this part a bit misleading as it jumps back and forth: things are inconclusive; a relation between pretest phonological awareness and intervention outcome is mentioned for which no statistic is provided in the results; two mentions that after p-value adjustment there were no effects left, but it is still discussed that these effects might suggest something. Especially at the start of the discussion there should be a clear summary of what was found first and then one can discuss what the presence and absence of effects might indicate.

We thank the reviewer for this comment and agree that the start of the discussion section is not as clear as it should be. We have amended the first paragraph to provide a better summary of the findings before proceeding on to our interpretation. Furthermore, we have amended the wording to make clear that the relationship between phonological awareness and age is related to the correlation post-hoc analysis in the results section. 

“Using a RCT design, we tested the hypotheses that struggling readers could leverage a phonemic image cue placed below the vowels in digitally presented text to improve reading accuracy for isolated words and connected text, and that this benefit would be more pronounced for those readers with lower performance on measures on phonological processing. Data collected after a two-week period of unsupervised (but digitally monitored) practice demonstrated that struggling readers could read more complex words using the tool: compared to the control group, the intervention group showed a significantly larger improvement in decoding accuracy specifically for pseudo-words. As depicted in the results, this benefit did not extend to real words or either measure related to connected-text reading (accuracy and rate). 

Although there was no benefit, stable performance on measures of connected text read was observed for all participants with no significant difference between groups. The lack of benefits for connected text reading might reflect the limited training period or the increased cognitive demands of a novel approach to reading. These are important questions for future studies as generalization to connected text is of key importance. 

Correlation analyses, moreover, were inconclusive but suggest that the tool may benefit those participants who are younger and/or have lower phonological processing scores (see Results)” 

P20:L16: [...] highly inconsistent grapheme-phoneme mapping "of English" [...]

The sentence now reads: “We focused on vowels because, in English, the highly inconsistent grapheme-phoneme mapping is a major hurdle for struggling readers.”

P22:L3-L5: Throughout the manuscript there are sentences which use many commas and therefore appear encapsulated and make it unnecessarily hard for the reader to grasp the most relevant information. In this case you can straight out write that there is an improved decoding of pseudowords and remove the subordinate clause. As this is the only comment regarding writing style, I also wanted to note that there appear to be a lot of double spaces in manuscript. Content wise it could be highlighted here that this improvement appeared after being trained with the app for (only) 2 weeks.

Thank you for this feedback - we certainly want to make sure that the manuscript is clear and readable. We have deleted the subordinate clause in that sentence and have removed double spaces. 

P23:L5-L13: This is an interesting observation. They appeared to have learned to make use of the cues, but this disappears in the testing situation. I feel it’s quite relevant to understand this better. Given that there is no performance penalty it seems like they might be able to ignore the cues altogether? One could take a more qualitative approach and investigate how children use it in these different scenarios or alternatively add an assessment mode into the app itself which blends with the intervention?

This is a great point and represents a significant hurdle for any pedagogy to motivate a challenged reader even when they know it works/helps. We had a constant debate about how to measure use while reading and will contend that it is possible the children were ignoring the symbols altogether. During testing the children were reminded quite frequently to utilize the symbols, but as we discuss in the manuscript: ‘Either due to the limited practice period, limited supervised practice, or conflict with existing strategies children use when approaching challenging words,’ they didn’t feel as motivated to adopt the cues. We agree that future studies should work to collect better measures to piece this apart and have added the following to that paragraph in the discussion:

“Future studies should incorporate qualitative and metacognitive methods to identify factors and circumstances that encourage struggling readers to adopt a novel strategy.” (P23, L22 - P24, L2). 

P23:L23: Given that parents also got a brief introduction to the app and you come back to it in the discussion it might be worth to recommend measuring parental involvement in future studies?

We agree. The paragraph now reads:

“Albeit promising, these results should be interpreted cautiously: Our power analysis indicated that we only had sufficient power to detect relatively large effects and many of the analyses were likely underpowered. Thus, future studies are needed with larger sample sizes to provide more conclusive results. Moreover, two additional points merit further investigation. First, future studies should more efficiently monitor practice adherence and cadence at home to better explore the relationship between exposure and reading-related measures. Second, given the short intervention period, we did not examine generalization to reading improvements without the cue and across different aspects of skilled reading. We only investigated whether the cue could be effectively used to decode more complex words. Thus, examining long-term learning effects and generalization to a variety of different contexts, as well as the role of parental involvement/participation is an important future direction.”

Summary:

In summary, the manuscript has much improved and will be a relevant addition to the field, but the provided analysis and code are not yet up to the required standards.

We thank the reviewer for these relevant and meaningful recommendations for our manuscript. We believe the work is now much stronger for it.

---

## [Decision Letter · Decision Letter 2]

19 Oct 2020

PONE-D-19-34461R2

Annotating digital text with phonemic cues to support decoding in struggling readers

PLOS ONE

Dear Dr. Donnelly,

Thank you for submitting your manuscript to PLOS ONE. After careful consideration, we feel that it has merit but does not fully meet PLOS ONE’s publication criteria as it currently stands. Therefore, we invite you to submit a revised version of the manuscript that addresses the points raised during the review process.

As you will see below, only Reviewer 2 suggests some further revisions.

We look forward to receiving your revised manuscript.

Kind regards,

Claudia Männel, PhD

Academic Editor

PLOS ONE

Reviewers' comments:

Reviewer's Responses to Questions

**Comments to the Author**

1. If the authors have adequately addressed your comments raised in a previous round of review and you feel that this manuscript is now acceptable for publication, you may indicate that here to bypass the “Comments to the Author” section, enter your conflict of interest statement in the “Confidential to Editor” section, and submit your "Accept" recommendation.

Reviewer #2: (No Response)

2. Is the manuscript technically sound, and do the data support the conclusions?

Reviewer #2: Partly

3. Has the statistical analysis been performed appropriately and rigorously? 

Reviewer #2: No

4. Have the authors made all data underlying the findings in their manuscript fully available?

Reviewer #2: Yes

5. Is the manuscript presented in an intelligible fashion and written in standard English?

Reviewer #2: Yes

6. Review Comments to the Author

Reviewer #2: I thank the authors once more for addressing the points I raised and providing an improved manuscript. A few minor points remain to be solved.

First, some comments relating to your response letter:

You mentioned that you provide violin plots as S1, but these were not visible in the resubmission. Furthermore, I would strongly suggest to replace the barplots in figures 1 and 2 with these violin plot.

In the last revision, I saw the document/section with the “transparent changes” but it was not part of this new resubmission and is neither in the repository (as is mentioned in the text) nor the pre-registration.

You wrote: "During testing the children were reminded quite frequently to utilize the symbols, but as we discuss in the manuscript: ‘Either due to the limited practice period, limited supervised practice, or conflict with existing strategies children use when approaching challenging words,’ they didn’t feel as motivated to adopt the cues."

Depending on whether these reminders happened during or in between testing trials this might be problematic, as it shifts the concept you measured away from the benefit of Sound It Out in a naturalistic setting, where children show what they learned, to one where adults instruct children what to do. This might also relate to the previously discussed finding that the children appeared to have learned to make use of the cues, but that this disappears in the testing situation. Can you clarify the exact procedure when children were “frequently reminded” to make use of the cues and add this to the method/discussion?

P14:L21: Could you clarify whether the randomization of the reading lists happened at the individual or intervention group level?

Please replace random effect with random intercept (as opposed to random slope, which are both a type of random effect):

P16:L11 a random intercept per participant

P16:L13 a random intercept for word list

P16:L17: this is not "post hoc" analysis

P17-P19: Thank your for explaining and improving the mixed models, which now seem appropriately done. The rest of the statistical analyses remain problematic though. First of all, only significant effects should be followed up with post-hoc tests. Otherwise, you run the risk of discovering spurious significant effects, and you have to adjust for multiple comparisons as well. The mixed model approach is much superior to t-tests because it takes into account the variance that can be attributed to test items and subjects in the random effects structure. In sum, the result of the model weight more than the t-tests, and if the model does not describe a significant interaction, this is the end of the analysis. If you wanted to base your discussion on the results of the t-tests, you would not have to run any models in the first place. If you want to do a post-hoc analysis of a mixed model you should furthermore not revert to t-tests but to approaches like least square means for multiple comparisons. Usually this is only done when you have factors with more than 2 levels and the pairwise comparisons cannot be read out of the model summary anymore (which does not apply to your model where group and time have 2 levels each). Instead, or on top, of plotting the raw data you might also want to plot the effects that the mixed model describes.

P19:L7 – In the response letter you indicated to replace prediction analysis with correlation analysis, but it still says prediction analyses here.

P21:L2-5: As mentioned in my last revision, while it is true that the intervention group had a bigger gain in pseudoword decoding than the control group, this has to be seen in the light of a significantly lower starting level. At post-test the two groups read equally well and from my perspective this should be the main result. Looking purely at increase from pre- to post-test is usually not relevant because both, regression to the mean as well as ceiling effects can explain your findings. All of these things are not discussed so far, and this is also why it is so important to provide better figures that tell the entire story.

Furthermore, if you want to draw a causal conclusion (i.e. that the game improves pseudoword decoding) the two intervention groups have to fulfil the requirement of exchangeability. Usually in an RCT this is given or assumed, but in small samples it often does not hold and cannot be corrected or controlled for, due to the limited number of covariates such a small regression analysis can afford. I also only noticed now that the randomization took place after the (pre-test) outcome measures had already been collected (P11:L22). So you were in the position to avoid this scenario and produce exchangeable groups during randomization (e.g. by matching). In any case, I would be very careful to draw causal conclusions in this scenario.

P21:L11-19. I still think this paragraph needs to be restructured as I suggested previously. First of all, it should be mentioned that there were no significant effects of the correlation analysis. Then you can discuss possible reasons (low sample size, multiple comparisons, etc.) and the trends you observed in the data at hand, before concluding that this is not enough evidence to support any firm claims.

In sum, I think the manuscript is on a good way. The introduction and method sections are ok. For the results I recommend removing the post-hoc t-tests and replacing the barplots with violin plots or model predictions. The discussion still has to be adjusted to reflect the actual findings, as some claims are still too strong and are not backed up by the data and the experimental design.

7. PLOS authors have the option to publish the peer review history of their article (what does this mean?). If published, this will include your full peer review and any attached files.

Reviewer #2: **Yes: **Toivo Glatz

---

## [Author Response · Author response to Decision Letter 2]

6 Nov 2020

We would like to thank the reviewers once again for their thoughtful and insightful comments on this study. We have made further revisions and agree with the reviewers’ points that these changes will make this manuscript suitable for publishing. The reviewers comments are below in black and our responses and specific actions are noted in blue.

Reviewer #2: I thank the authors once more for addressing the points I raised and providing an improved manuscript. A few minor points remain to be solved.

First, some comments relating to your response letter:

You mentioned that you provide violin plots as S1, but these were not visible in the resubmission. Furthermore, I would strongly suggest to replace the barplots in figures 1 and 2 with these violin plot.

We apologize that this figure was not readily visible to you. In line with the PLOS ONE guidelines, it was in the zip file of Supplemental Information. We agree that the violin plots are a very important addition to transparency in our data visualizations. We have amended figures 1 and 2 to now include these violin plots in addition to the bar plots. The figure also now displays differences scores for each individual as lines on the violin plots. 

In the last revision, I saw the document/section with the “transparent changes” but it was not part of this new resubmission and is neither in the repository (as is mentioned in the text) nor the pre-registration.

We apologize for this oversight. The document can be found in our pre registration file repository, here: https://osf.io/cy5ms/. 

You wrote: "During testing the children were reminded quite frequently to utilize the symbols, but as we discuss in the manuscript: ‘Either due to the limited practice period, limited supervised practice, or conflict with existing strategies children use when approaching challenging words,’ they didn’t feel as motivated to adopt the cues."

Depending on whether these reminders happened during or in between testing trials this might be problematic, as it shifts the concept you measured away from the benefit of Sound It Out in a naturalistic setting, where children show what they learned, to one where adults instruct children what to do. This might also relate to the previously discussed finding that the children appeared to have learned to make use of the cues, but that this disappears in the testing situation. Can you clarify the exact procedure when children were “frequently reminded” to make use of the cues and add this to the method/discussion?

We apologize that our previous explanation was ambiguous and agree that frequent reminders and researcher-motivations would impact our study’s findings and interpretations. To clarify, at the start of each testing session children were reminded: (1) that they were not being timed so they could take their time with each word to be as accurate as possible and (2) that the symbols were there to help them should they come to a challenging word.

The following has been added to the methods section:

“At the start of administration, all participants were reminded that they were not being time and encouraged to read as accurately as possible. For intervention participants exposed to the image cue, participants were additionally reminded that the symbols were there to help them should they come to a challenging word.” (p15, 2-5) 

 “As with the decoding measures, participants were reminded at the start of administration that they were not being timed, encouraged to read as accurately as possible, and (for intervention participants) that the symbols were there should they come to a challenging word.” (p 15, 18-21) 

P14:L21: Could you clarify whether the randomization of the reading lists happened at the individual or intervention group level?

Randomization of reading lists happened at the individual level. 

Please replace random effect with random intercept (as opposed to random slope, which are both a type of random effect):

P16:L11 a random intercept per participant

P16:L13 a random intercept for word list

We have updated the terminology accordingly. 

P16:L17: this is not "post hoc" analysis

As these correlation analyses weren’t planned and we considered them an interesting extension of the observed data, we feel that they satisfy the definition of post-hoc (we also realize that there are a variety of definitions of “post-hoc”). To make clear that they were performed with limited data, we have revised the manuscript to reflect that these are considered exploratory in nature. 

P17-P19: Thank your for explaining and improving the mixed models, which now seem appropriately done. The rest of the statistical analyses remain problematic though. First of all, only significant effects should be followed up with post-hoc tests. Otherwise, you run the risk of discovering spurious significant effects, and you have to adjust for multiple comparisons as well. The mixed model approach is much superior to t-tests because it takes into account the variance that can be attributed to test items and subjects in the random effects structure. In sum, the result of the model weight more than the t-tests, and if the model does not describe a significant interaction, this is the end of the analysis. If you wanted to base your discussion on the results of the t-tests, you would not have to run any models in the first place. If you want to do a post-hoc analysis of a mixed model you should furthermore not revert to t-tests but to approaches like least square means for multiple comparisons. Usually this is only done when you have factors with more than 2 levels and the pairwise comparisons cannot be read out of the model summary anymore (which does not apply to your model where group and time have 2 levels each). Instead, or on top, of plotting the raw data you might also want to plot the effects that the mixed model describes.

Although we do not agree with the reviewers points here, we have, nonetheless, removed the t-statistics from the results section. It is our opinion that many readers will be left wondering about those analyses and that they provide an interesting, contextual look into the results that are couched within the interaction effects. The reason that we had included those statistics is in response to specific requests from others that have given feedback on this work. But, once again, those statistics are not critical to the paper and we have removed them.

P19:L7 – In the response letter you indicated to replace prediction analysis with correlation analysis, but it still says prediction analyses here.

Thanks for catching this - have amended accordingly.

P21:L2-5: As mentioned in my last revision, while it is true that the intervention group had a bigger gain in pseudoword decoding than the control group, this has to be seen in the light of a significantly lower starting level. At post-test the two groups read equally well and from my perspective this should be the main result. Looking purely at increase from pre- to post-test is usually not relevant because both, regression to the mean as well as ceiling effects can explain your findings. All of these things are not discussed so far, and this is also why it is so important to provide better figures that tell the entire story.

Furthermore, if you want to draw a causal conclusion (i.e. that the game improves pseudoword decoding) the two intervention groups have to fulfil the requirement of exchangeability. Usually in an RCT this is given or assumed, but in small samples it often does not hold and cannot be corrected or controlled for, due to the limited number of covariates such a small regression analysis can afford. I also only noticed now that the randomization took place after the (pre-test) outcome measures had already been collected (P11:L22). So you were in the position to avoid this scenario and produce exchangeable groups during randomization (e.g. by matching). In any case, I would be very careful to draw causal conclusions in this scenario.

We thank the reviewer for this comment and agree that care needs to be taken to ensure that our interpretations are supported by our results. This study represents a small-scale ‘proof of concept’ study with encouraging results, but we take care to describe our findings without use of causal language. As regression to the mean and ceiling effects are important factors to keep in mind, the following has been adjusted in the discussion:

“Albeit promising, these results should be interpreted cautiously: Our power analysis indicated that we only had sufficient power to detect relatively large effects and many of the analyses (e.g., individual differences) were likely underpowered. Also, as there was a significant difference at pre-test for our sole finding with pseudo word decoding, future studies are needed to rule out the role of regression toward the mean and possible ceiling effects. Thus, future studies are needed with larger sample sizes to provide more conclusive results.” (p24.,10-12) 

As to randomization, the participants were randomly assigned to groups prior to participation in the pre-test for the study. This is explained in the sentence following the one you reference: “Randomization was unconstrained with group assignment determined at time of consent”

We understand that the language in the Study Design section is ambiguous - the intention there was to make it clear that all participants had identical pre-testing prior to deviating in study protocol for their assigned group. We have amended that section to instead read:

“In a randomized pre-post design, participants were randomized to a control or intervention condition. Randomization was unconstrained with group assignment determined at time of consent; however, sibling participants were assigned to the same group to better control participant adherence. Both intervention and control participants completed an initial, baseline session that collected all outcome measures using the normal text condition presented on a Kindle fire tablet without the Sound It Out cue (see Outcome Measures).” (p11-12, 20-2) 

P21:L11-19. I still think this paragraph needs to be restructured as I suggested previously. First of all, it should be mentioned that there were no significant effects of the correlation analysis. Then you can discuss possible reasons (low sample size, multiple comparisons, etc.) and the trends you observed in the data at hand, before concluding that this is not enough evidence to support any firm claims.

For clarity with our interpretations, we have amended that paragraph to read:

“Correlation analyses, after multiple comparison correction, revealed no significant relationships between our variables of interest and benefit of the cue. Due to unreliable practice data (see Statistics), analyses cannot support any conclusions regarding the relationship between subject characteristics and benefits conferred by phonemic cues. However, results suggest that the tool may benefit those participants who are younger and/or have lower phonological processing scores (see Results). Together, although most analyses failed to meet our adjusted significance threshold, data suggests that participants were able to effectively use the cues in isolated situations (i.e. pseudoword reading), but the tool did not become sufficiently automatic to produce significant gains in passage reading fluency.” (p.21-2, 16-2)

In sum, I think the manuscript is on a good way. The introduction and method sections are ok. For the results I recommend removing the post-hoc t-tests and replacing the barplots with violin plots or model predictions. The discussion still has to be adjusted to reflect the actual findings, as some claims are still too strong and are not backed up by the data and the experimental design.

We greatly appreciate the reviewer’s support and commitment to making this work a more impactful contribution to the literature.

---

## [Decision Letter · Decision Letter 3]

20 Nov 2020

PONE-D-19-34461R3

Annotating digital text with phonemic cues to support decoding in struggling readers

PLOS ONE

Dear Dr. Donnelly,

Thank you for submitting your revised manuscript to PLOS ONE. We invite you to submit a revised version of the manuscript that incorporates the final minor changes requested by Reviewer 2. I will then be happy to accept your manuscript for publication.

We look forward to receiving your revised manuscript.

Kind regards,

Claudia Männel, PhD

Academic Editor

PLOS ONE

Reviewers' comments:

Reviewer's Responses to Questions

**Comments to the Author**

1. If the authors have adequately addressed your comments raised in a previous round of review and you feel that this manuscript is now acceptable for publication, you may indicate that here to bypass the “Comments to the Author” section, enter your conflict of interest statement in the “Confidential to Editor” section, and submit your "Accept" recommendation.

Reviewer #2: (No Response)

2. Is the manuscript technically sound, and do the data support the conclusions?

Reviewer #2: Yes

3. Has the statistical analysis been performed appropriately and rigorously? 

Reviewer #2: Yes

4. Have the authors made all data underlying the findings in their manuscript fully available?

Reviewer #2: Yes

5. Is the manuscript presented in an intelligible fashion and written in standard English?

Reviewer #2: Yes

6. Review Comments to the Author

Reviewer #2: Thank you for your detailed answers and providing a, once more, much improved manuscript. I would suggest incorporating the following minor changes prior to publication:

You now mention the differences in pseudoword decoding at pre-test in the discussion, but these are not yet reported in the results section. Please add the relevant statistics to the results section, if possible with effect size.

I would suggest to add "small scale RCT" (on P21:L11) and/or "small scale proof of concept" (on P26:L17).

P24:L2-5: The repeated use of "future studies" reads a bit awkward. I'd suggest to slightly rephrase.

7. PLOS authors have the option to publish the peer review history of their article (what does this mean?). If published, this will include your full peer review and any attached files.

Reviewer #2: **Yes: **Toivo Glatz

---

## [Author Response · Author response to Decision Letter 3]

20 Nov 2020

We would like to thank the reviewer and have provided responses for each concern. The reviewers comments are below in black and our responses and specific actions are noted in blue.

Reviewer #2: Thank you for your detailed answers and providing a, once more, much improved manuscript. I would suggest incorporating the following minor changes prior to publication:

You now mention the differences in pseudoword decoding at pre-test in the discussion, but these are not yet reported in the results section. Please add the relevant statistics to the results section, if possible with effect size.

The results has been added and now reads:

“At pretest, despite randomization, the intervention group by chance had lower scores than the control group: this is evidenced by the significant main effect of group in the mixed effects model (β=-6.2, t(156) = -2.65, p=0.009).” 

I would suggest to add "small scale RCT" (on P21:L11) and/or "small scale proof of concept" (on P26:L17).

We have amended the text to reflect your suggestions. Those lines are now amended to read:

“Using a small scale RCT design, we tested the hypotheses that …”, and “In aggregate, these findings represent a small scale proof-of-concept for this …”

P24:L2-5: The repeated use of "future studies" reads a bit awkward. I'd suggest to slightly rephrase.

We agree and have amended to the following:

“Also, as there was a significant difference at pre-test for our sole finding with pseudo word decoding, future studies are needed to rule out the role of regression toward the mean and possible ceiling effects. Thus, future work is needed with larger sample sizes to provide more conclusive results. Moreover, two additional points merit further investigation. First, future experiments should more efficiently, and quantitatively and qualitatively, monitor practice adherence and cadence at home to better explore the relationship between exposure and reading-related measures.”

---

## [Editor Report · Decision Letter 4]

23 Nov 2020

Annotating digital text with phonemic cues to support decoding in struggling readers

PONE-D-19-34461R4

Dear Dr. Donnelly,

We’re pleased to inform you that your manuscript has been judged scientifically suitable for publication and will be formally accepted for publication once it meets all outstanding technical requirements.

Kind regards,

Claudia Männel, PhD

Academic Editor

PLOS ONE
---

## [Editor Report · Acceptance letter]

25 Nov 2020

PONE-D-19-34461R4 

Annotating digital text with phonemic cues to support decoding in struggling readers 

Dear Dr. Donnelly:

I'm pleased to inform you that your manuscript has been deemed suitable for publication in PLOS ONE. Congratulations! Your manuscript is now with our production department. 

Kind regards, 

on behalf of

Dr. Claudia Männel 

Academic Editor

PLOS ONE